# Tridimensional infiltration of DNA viruses into the host genome shows preferential contact with active chromatin

Pierrick Moreau [1,2,3], Axel Cournac[4,5], Gianna Aurora Palumbo[1,2,3], Martial Marbouty[4,5], Shogofa Mortaza[4,5], Agnes Thierry[4,5], Stefano Cairo[6], Marc Lavigne[3,7], Romain Koszul [4,5] & Christine Neuveut [1,2,3]

Whether non-integrated viral DNAs distribute randomly or target specific positions within the higher-order architecture of mammalian genomes remains largely unknown. Here we use Hi-C and viral DNA capture (CHi-C) in primary human hepatocytes infected by either hepatitis B virus (HBV) or adenovirus type 5 (Ad5) virus to show that they adopt different strategies in their respective positioning at active chromatin. HBV contacts preferentially CpG islands (CGIs) enriched in Cfp1 a factor required for its transcription. These CGIs are often associated with highly expressed genes (HEG) and genes deregulated during infection. Ad5 DNA interacts preferentially with transcription start sites (TSSs) and enhancers of HEG, as well as genes upregulated during infection. These results show that DNA viruses use different strategies to infiltrate genomic 3D networks and target specific regions. This targeting may facilitate the recruitment of transcription factors necessary for their own replication and contribute to the deregulation of cellular gene expression.

[1] Institut Pasteur, Unité Hepacivirus et Immunité Innée, 75015 Paris, France. [2] CNRS, UMR 3569, 75015 Paris, France. [3] Institut Pasteur, Département de Virologie, Paris, France. [4] Institut Pasteur, Département Génomes et Génétique, Groupe Régulation spatiale des génomes, 75015 Paris, France. [5] CNRS, UMR 3525, 75015 Paris, France. [6] XenTech, Research and Development Department, 91000 Evry, France. [7] Institut Cochin-INSERM U1016-CNRS UMR8104, Université Paris Descartes, Paris, France. These authors contributed equally: Pierrick Moreau, Axel Cournac. Correspondence and requests for materials should be addressed to R.K. (email: romain.koszul@pasteur.fr) or to C.N. (email: christine.neuveut@pasteur.fr)

The three-dimensional organization of the genome has emerged in recent years as an important player in gene expression regulation. The cellular genome is compacted through folding of the DNA fiber that is organized into sub-Mb chromosomal "topologically associated domains" (TADs), i.e., DNA regions that display increased contact over themselves when analyzed through chromosome conformation capture experiments (Hi-C)[1–3]. Chromatin loops resulting from the contacts between discrete, distant loci can notably result in functional interactions between co-regulated genes or enhancers and promoters. TADs are themselves segregated spatially in compartment A and B depending on their gene content (i.e., gene-rich and transcriptionally active vs. gene-poor and transcriptionally silent, respectively).

Viruses that transcribe their DNA within the nucleus have to adapt to the molecular mechanisms that govern transcriptional regulation. They can modify and/or re-direct the host's transcriptional machinery in order to induce viral gene expression, as documented and supported by numerous studies[4]. However, the appreciation of the complexity of the higher-order organization of the host genome and its potential influence in the regulation of gene expression raises questions regarding the spatial arrangement of non-integrated viral DNA in this context. Notably, whether the overall distribution of the viral DNA will be random, or not.

To investigate this question, we focused on two DNA viruses, hepatitis B virus (HBV) and Human adenovirus. Chronic HBV infection is the prevalent cause of liver cancer. HBV is a small enveloped DNA virus of 3.2 kb which replicates its genome in the cytoplasm via reverse transcription of the encapsidated pre-genomic RNA (pgRNA) into a relaxed circular partially double-stranded DNA (RC-DNA). Upon internalization and transfer to the nucleus, the RC-DNA is converted into covalently closed circular DNA (cccDNA)[5]. cccDNA is a template for all HBV transcripts including pgRNA and is organized into a chromatin-like structure associated with histone and non-histone proteins[6]. Its transcription depends on cellular transcription factors (TFs) and chromatin-modifying enzymes, as well as on the viral regulatory protein HBx[7]. Human adenovirus is part of a large family of viruses that infect a wide range of vertebrate hosts. Adenoviruses infect and replicate at various sites of the respiratory tract causing respiratory diseases, but it can also infect the eyes and gastrointestinal tract as well as, albeit less frequently, urinary bladder and liver. Adenovirus is a non-enveloped virus containing a linear double-stranded DNA genome of ~40 kb. Upon infection, viral DNA is delivered to the nucleus, where it associates with histones and viral protein VII[8]. Adenovirus transcription is by convention separated into early and late phases separated by viral DNA replication[8].

Here, we applied Hi-C[9] and viral DNA capture (CHi-C) on primary human hepatocytes (PHH) infected by HBV or Hi-C on PHH infected by adenovirus type 5 (Ad5) to investigate the spatial nuclear organization of the viral DNAs. We showed that while both viral DNAs contact preferentially active chromatin, they adopt different strategies. HBV DNA contacts preferentially CpG islands (CGIs) that are enriched for Cfp1 factors, consistent with our finding that Cfp1 is recruited on the viral cccDNA and is required for H3K4me3 and full HBV transcription. Moreover we observed that CGIs contacted by HBV are enriched for CGIs associated with highly expressed genes and also associated with deregulated genes in HBV-infected PHH. Finally, we demonstrated that Ad5 interacts preferentially with transcription start sites (TSSs) and enhancers of highly expressed genes and genes upregulated during infection.

## Results

### Genome organization of HBV-infected and non-infected PHH.
We first investigated the response to HBV infection of PHH from two independent donors (399 and 342) from the viewpoint of their genome organization. Hi-C libraries of PHH cells harvested 7 days after infection when cccDNA level is maximal[10] (Fig.1a, and Supplementary Fig.1) were generated and contact maps of the host and viral genomes were computed (Supplementary Table 1; Methods)[9,11,12]. TADs and compartments[3,9] were called at 100 kb resolution using HiCSeg and eigen vector decomposition, respectively (Supplementary Fig. 2). A 400 kb binning to characterize contacts between the host and viral genome was selected as a tradeoff between the hierarchical higher-order chromatin structures and the distribution of contacts made by the small viral genome. At this resolution, the organization of the genome of hepatocytes appears similar whether the cells are infected or not (Fig. 1b; Supplementary Figs 3, 4 and 5). At 100 kb resolution, the PHH genome organization is also globally similar to those of other cell types regarding TADs organization (Supplementary Fig. 2d). Note that fine changes may nevertheless go unnoticed at this resolution. The contact frequencies as a function of genomic distance p($s$) were also similar in PHH and fibroblasts[12], as well as in non-infected and infected PHH (Supplementary Fig. 5c), suggesting that overall chromatin conformation is not altered during infection. Eigen vector decomposition and RNA-seq performed in infected and non-infected PHH 399 partitioned the genome into compartments A (active) and B (inactive) (Fig. 1c, d, Methods), a separation also supported by PHH histone mark H3K4me3 deposition[13] (Fig. 1e). Overall, PHH contact maps generated using independent donors show that A and B compartments remain unchanged during infection (Fig. 2a), which does not induce large-scale reorganization of hepatocyte genomes.

### HBV contacts the host genome preferentially at CpG islands.
To enrich the libraries in HBV-human DNA contacts, a capture approach using biotinylated oligonucleotides covering the viral genome was implemented (Methods)[13,14]. A total of 39,077 and 30,696 specific contacts between the host and the viral genome were characterized for donors 399 and 342, respectively (Supplementary Table 1). The HBV genome makes contacts with the entire host genome (Fig. 1f), with a significant preference for compartment A (Fig. 2b) and active chromatin regions enriched in H3K4me3, H3K4me2, as well as H3K27ac marks[15] (Fig. 2c). Preferential contacts between the viral and host genomes were investigated for specific functional genomic groups (Fig. 2a and Supplementary Fig. 6). No preferential association of HBV genome with published HBV integration sites[16] was observed. This result is consistent with the hypothesis that cccDNA is not the precursor of integration[17]. Contact frequencies were significantly depleted between the virus and lamin B1-associated domains, i.e., regions corresponding predominantly to B compartments[18] (Fig. 3a and Supplementary Fig. 6). A significant enrichment was observed for TSS and CpG islands (CGIs), which are short interspersed DNA sequences positioned within most (70%) promoters and that controls gene expression through chromatin regulation (Fig. 3a, Supplementary Figs 6 and 7a)[19]. To investigate the link between viral contact enrichment at CGIs and TSS, CGIs were separated in three groups according to their distance to the nearest TSS. Strong enrichment in HBV contacts were recovered for CGIs overlapping or positioned within 0–50 kb of a TSS (Fig. 3b), i.e., close to promoters. These CGIs were also enriched in PolII and H3K4me3 deposition[15] (Supplementary Fig. 7b), showing that HBV contacts CpG islands upstream of active genes. Functional annotation analysis of genes in the vicinity of CGIs contacted by the viral genome showed enrichment of homeobox family genes, suggesting potential deregulation of cell identity and metabolism[20,21] (Supplementary Data 1).

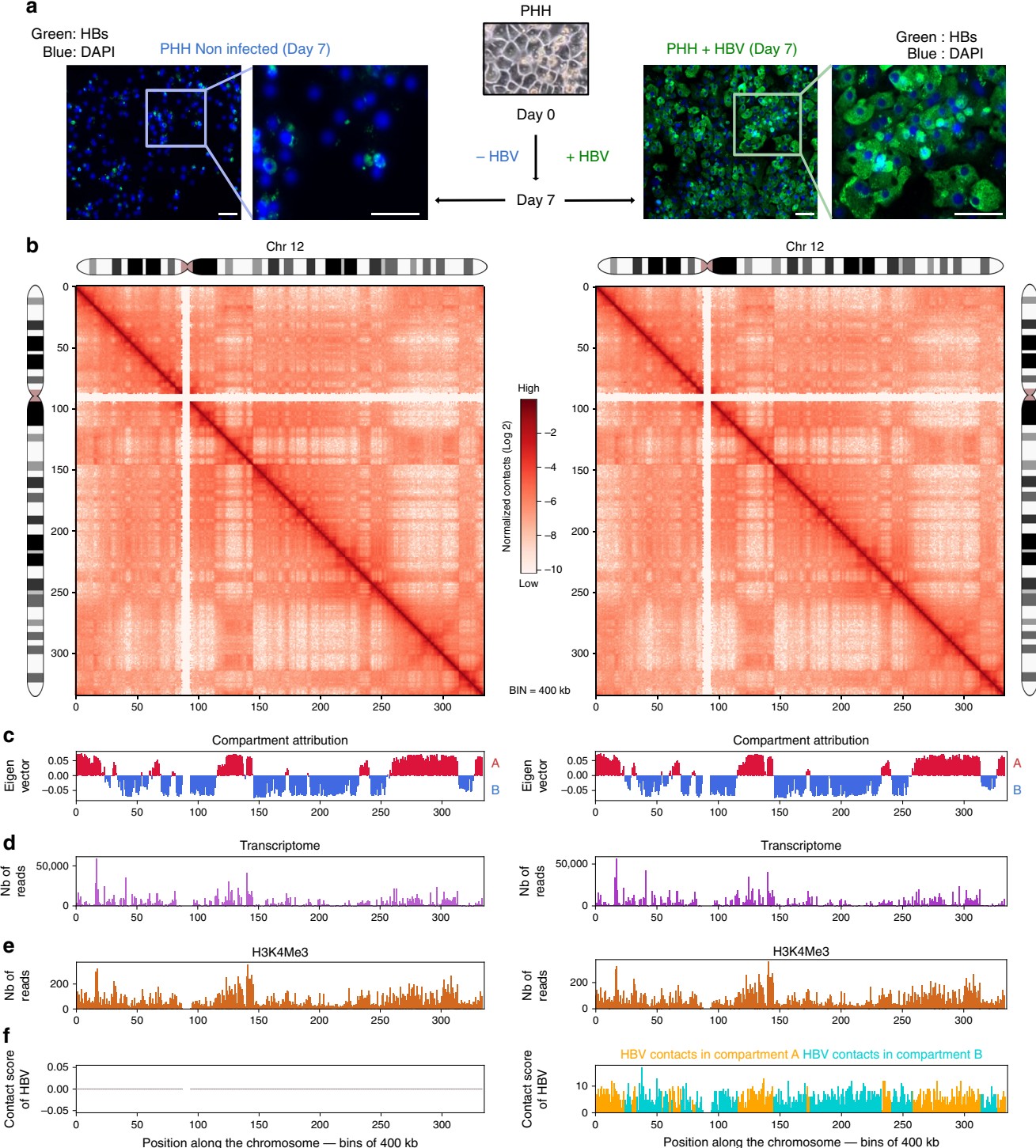

**Fig. 1** Distribution of HBV contacts along the host genome. **a** HBV envelope proteins (HBs, green) were visualized using immunofluorescence in PHH infected or not by HBV. Scale bar represents 50 μm. **b** Hi-C contact maps of chromosome 12 from non-infected PHH (left) or from PHH infected with HBV (right) at 400 kb resolution. The color scale represents the normalized contact frequencies between bins. **c** First principal component of chromosome 12 (400 kb resolution) representing the active A-type (red) and repressed B-type compartmentalization (blue). **d** Gene expression in non-infected and infected PHH 399 using RNAseq reads. **e** Active histone mark (H3K4me3) reads density along chromosome 12 (400 kb resolution) in HBV-infected PHH from Tropberger et al.[13]. **f** Distribution of HBV contacts along chromosome 12 (400 kb resolution) in infected PHH (right). Contacts with active chromatin (A) are shown in orange and with repressed chromatin (B-type) in green

**Impact of HBV DNA contacts on cellular gene expression**. We next assessed whether HBV and CGIs contacts influence infected and non-infected PHH gene expression (Methods). A total of 1139 mRNA (609 down- and 530 upregulated) and 31 LncRNA (15 up- and 14 downregulated) were differentially expressed

between the two conditions ($q$ value < 0.05) (Fig. 4a, Supplementary Table 2, Supplementary Data 2, 3, 4 and 5). KEGG pathway enrichment analysis showed significant enrichment in metabolic and liver-associated pathways such as complement and coagulation cascade, fructose and mannose metabolism, or

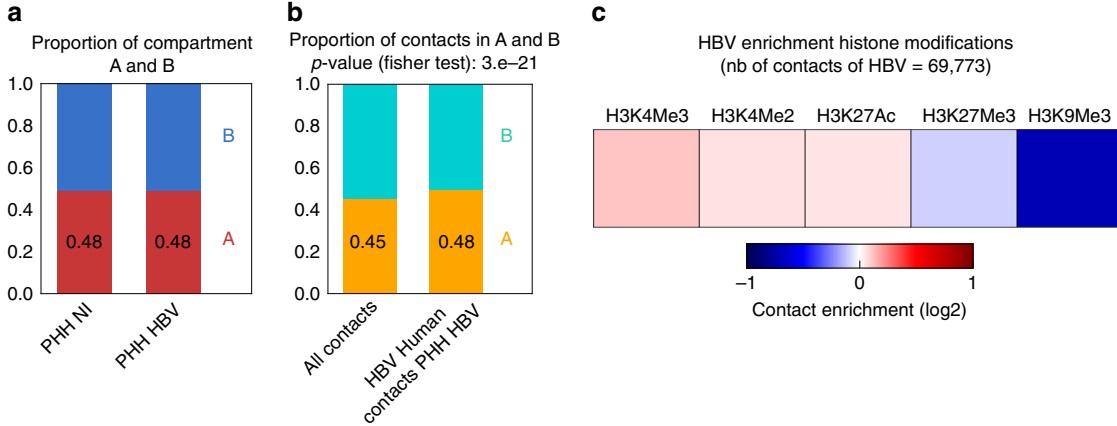

**Fig. 2** HBV genome contacts are enriched at active chromatin. **a** Distribution of active A and inactive B compartments in non-infected and infected PHH (400 kb resolution). **b** Distribution of Hi-C reads coverage in compartments A and B and distribution of HBV reads contacts on the genome in infected PHH (400 kb resolution) (statistical analysis was performed using Fisher's test). **c** Heatmap of HBV contacts enrichment for different histone modification marks (computed in windows ± 3.5 kb around the start of the read; N realizations = 100). Signals for the heatmap are represented by fold change (log2 ratio) compared to the average of random groups (see Methods). Histone modification marks dataset were from the Encode project generated using hepatic HepG2 cells (Encode Project:GSE29611)[15]

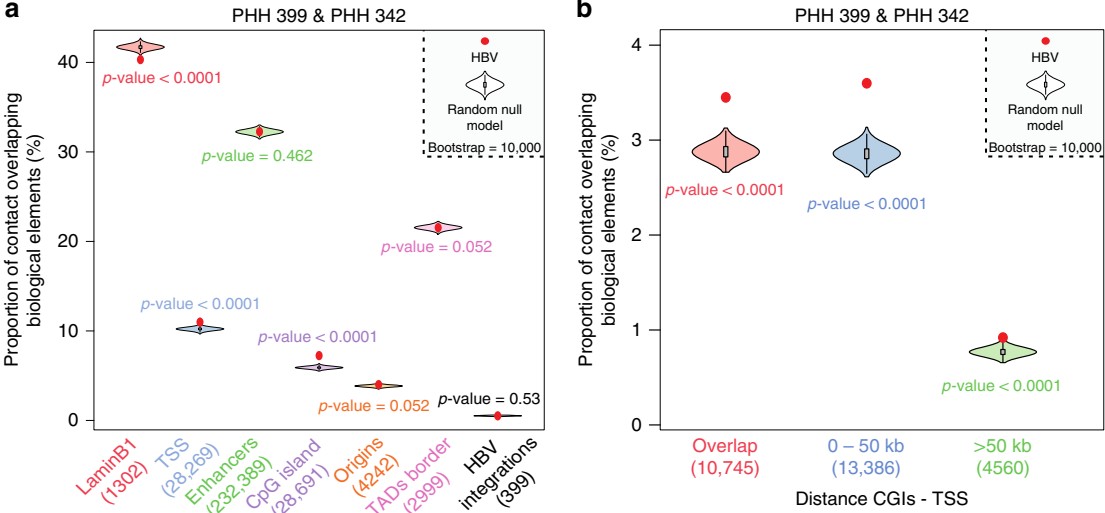

**Fig. 3** HBV genome contacts preferentially CpG-rich regions. **a** Proportion of HBV contacts with a given biological element (as indicated along with the number of elements) compared with the corresponding null model that shows the distribution using the Hi-C coverage of the corresponding library and represented by violin plot. Density plot width = frequency, line = 95% confidence interval, box plot = interquartile range. Red dot represents the proportion of HBV contacts that overlap with the indicated biological element (computed in windows ± 3.5 kb around the start of the read). p-values were determined according to the percentages detected in the bootstrap strategy (see Methods). **b** Proportion of HBV contacts with CGIs grouped according to their relative distance to TSS (computed in windows ± 3.5 kb around the start of the read) compared with the corresponding null model that shows the distribution using the Hi-C coverage and represented using the violin plot. Density plot width = frequency, line = 95% confidence interval, box plot = interquartile range. Red dot represents the proportion of HBV contacts that overlap with the indicated biological element (computed in windows ± 3.5 kb around the start of the read). p-values were determined according to the percentages detected in the bootstrap strategy (see Methods)

non-alcoholic fatty liver disease (Fig. 4a). Contacts were significantly enriched at CGIs associated with highly expressed genes, supporting our finding showing enrichment at active chromatin. HBV genomes contacts were also enriched at CGIs associated with deregulated genes compared with other CGIs categories (Fig. 4b), suggesting that HBV may interfere with gene expression through DNA/DNA contacts. Using RT-qPCR, we confirmed the modulation of the expression of genes in the vicinity of HBV contacted CGIs and differentially expressed in infected PHH (Supplementary Fig. 7c).

**CGIs-enriched Cfp1 is required for HBV transcription**. We next asked what could be the benefit for the virus to contact CGIs. CGIs generally lack DNA methylation[19,22]. The cellular factor CXXC finger protein 1 (CXXC1 or Cfp1) binds to non-methylated CGIs, establishing an active chromatin state in part through the recruitment of the methyltransferase Set1 responsible for H3K4me3 deposition[22,23]. Interestingly, DNA methylation as well as Set1-mediated H3K4 methylation of the viral cccDNA influence HBV transcription[24–26]. We silenced Cfp1 in HepG2-NTCP cells previously infected with HBV, as confirmed by

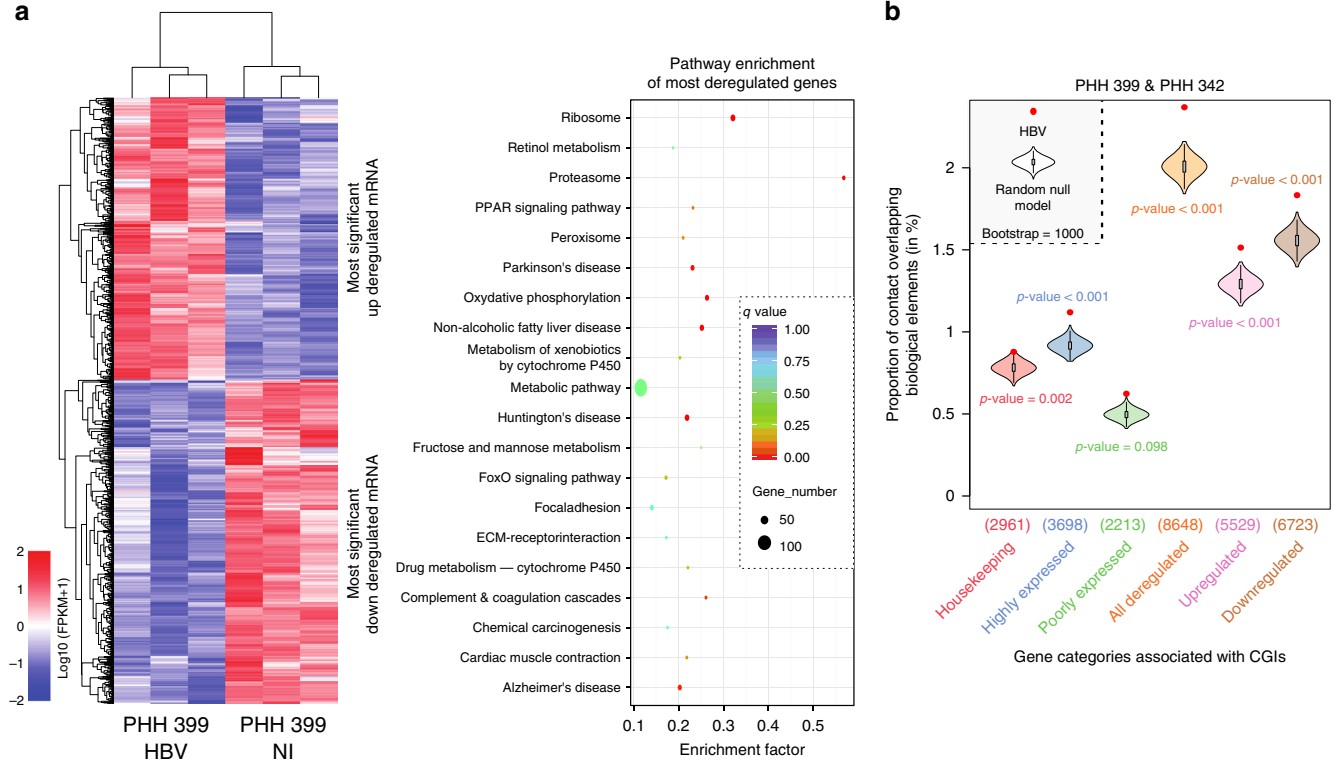

**Fig. 4** HBV contacts are enriched at GCIs associated with highly expressed genes and deregulated genes. **a** Heatmap of downregulated and upregulated genes in HBV infected PHH 399 compared with mock-infected PHH 399 and KEGG enrichment analysis. **b** Proportion of HBV contacts that overlap with CGIs present in the promoter of genes pooled according to their functional annotation (as indicated along with the number of elements) compared with the corresponding null model that shows the distribution using the Hi-C coverage and represented by violin plot. Density plot width = frequency, line = 95% confidence interval, box plot = interquartile range. Red dots represent the proportion of HBV contacts with the indicated biological element (computed in windows ± 3.5 kb around the start of the read). *p*-values were determined according to the percentages detected in the bootstrap strategy (see Methods)

RT-qPCR and western blotting (Fig. 5a; Methods). Depletion of Cfp1 decreased HBV RNA (Fig. 5a), but not cccDNA levels (Supplementary Fig. 8). These results suggest that Cfp1 is required for HBV transcription. We next investigated the functional relevance of Cfp1 for the transcription of cccDNA by ChIP-qPCR, and showed that Cfp1 is recruited directly onto the cccDNA (Fig. 5b). In addition, depletion of Cfp1 correlates with a decrease of H3K4me3 marks along the cccDNA (Fig. 5b). Altogether these results suggest that Cfp1 regulates HBV transcription through the recruitment of H3K4 methyltransferase allowing the deposition of H3K4me3 marks, and that HBV contacts preferential regions in the host genome that are enriched in factors important for its own transcription, such as Cfp1.

**HBx protein is not involved in HBV DNA targeting at CGIs.** Viral replication in vivo requires the regulatory protein HBx that acts mainly at the level of HBV transcriptional activation[10,27]. Interestingly HBx is recruited on both cccDNA and cellular promoters[10,28–31]. We thus studied whether HBx regulates HBV DNA/cellular genome contacts. To this aim, Hi-C libraries of PHH 399 infected with HBV deficient for HBx expression (HBV X-) were generated (Supplementary Fig. 9 and Supplementary Table 1). The contact maps of the host and viral genomes show that the organization of the hepatocytes genome is similar whether the cells are infected with HBV X- or HBV wild type (compare Fig. 6a and Supplementary Figs 10, 11a to Fig. 1b and Supplementary Figs 3, 4 and 5a). The contacts frequencies as a function of genomic distance p(s) for non-infected PHH, and PHH infected with HBV wild type or HBV X-, were also similar

(Supplementary Fig. 11b), suggesting that overall chromatin conformation is not altered for these conditions. Compartments A and B were characterized using the eigen vector decomposition as described above (Fig. 6b, c, Methods). A capture-step was performed to enrich the library in HBV sequences, resulting in 116,502 specific contacts between the host and the HBV X-genome (Supplementary Table 1; Methods). HBV X- makes contacts with the entire genome with a preference for compartment A (Fig. 6e). Similarly to HBV wild type, HBV X- is depleted from lamin B1 regions, but enriched at CpG islands (Fig. 6f). All together our results suggest that HBx does not drive specific contacts with the host chromatin.

**Adenovirus infection of hepatocytes: an alternative strategy.** An additional PHH (345) was derived from a 7-month-old donor infected by the human adenovirus serotype 5 (Ad5). To better understand the interplay between the Ad5 and PHH genome, we investigated using Hi-C, the positioning of Ad5 genome in the host nucleus, during infection (Methods) (Fig. 7a and Supplementary Fig. 12).

Hi-C contact maps were generated 4 and 7 days post plating (p. p.) when Ad5 replication is low and high, respectively (Fig. 7b Supplementary Figs 13 and 14a). PHH histone mark H3K4me3 deposition[13], and eigen vector decomposition partitioned the genome into compartments A (active) and B (inactive) (Fig. 7c, d, Methods). TADs positioning and compartments distribution were conserved when comparing non-infected PHH (used for HBV experiments) and infected PHH at day 4 (Fig. 7f, and Supplementary Fig. 14b). At day 7, a reorganization of TADs was

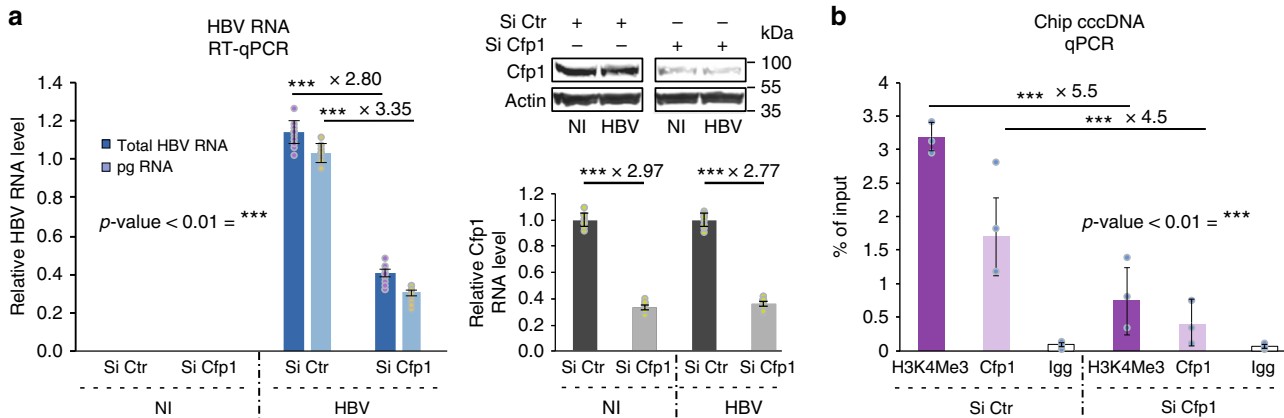

**Fig. 5** Cfp1 is required for HBV transcription. **a** HBV transcription was analyzed by RT-qPCR in HepG2 NTCP infected or not with HBV and transfected with the indicated siRNA (left graph). The expression of Cfp1 was analyzed by RT-qPCR and western blotting (lower right and upper right panel, respectively). Error bars of nine independent experiments ($n = 9$) represent standard error of the mean (SEM). ***$p < 0.01$ by Wilcoxon–Mann–Whitney test. **b** HepG2-NTCP cells were infected by HBV and Cfp1 expression was silenced or not as in **a**. Cfp1 and H3K4me3 recruitment was analyzed by ChIP-qPCR. Error bars ($n = 3$) represent SEM. ***$p < 0.01$ by Wilcoxon–Mann–Whitney test

detected (Supplementary Fig. 14b), corresponding to a shift of domains from the B to the A compartment (Fig. 7d, f and Supplementary Fig. 14c, d). The contact frequency as a function of genomic distance was conserved between different conditions, suggesting that Ad5 infection impacts TADs distribution without dramatically modifying the local chromatin structure (Supplementary Fig. 14e). Interestingly, Ad5 infection of quiescent fibroblasts induces a deep reprogramming of cellular gene expression with notably the activation of genes involved in cell proliferation and in chromatin regulation[32]. Part of these modulations can be attributed to Ad5 E1A gene products that can modulate the activity of proteins, such as the Rb family or Med23, from the mediator complex[32–35].

**Ad5 contacts preferentially active compartments of the host genome**. We identified 3862 and 1,352,760 unique viral–host contacts (Supplementary Table 1) at day 4 and 7 p.p., respectively, which significantly involved active chromatin regions (Fig. 7e, g). In agreement with this observation, Ad5 contacted preferential regions enriched for active marks known to be present at active promoters and enhancers (Fig. 7h). Intriguingly, a small enrichment between Ad5 and regions displaying the repressive H3K27me3 mark was also observed at day 7 p.p., but not 4 p.p. (Fig. 7h), raising the possibility that those regions may have shifted to compartment A, a hypothesis which remains to be tested. Ad5 genomes at day 4 and 7 p.p. were depleted from lamin B1 regions, but enriched at TSS, enhancers, CpG islands, and TADs border in the PHH (Fig. 8a). The enrichment of Ad5 contacts at TADs border may reflect the enrichment of enhancers at this localization. Enrichment plot analysis also confirmed that Ad5 preferentially contacts TSS, CGIs, and enhancers in PHH harvested 7 days p.p (Fig. 8b), in agreement with preferred contacts with regions enriched in active histone marks (Fig. 7h). We next looked whether TSS and enhancers contacted by Ad5 were enriched for specific transcription factor binding motifs. HOMER analysis showed that motifs for FOXA1, FOXA2, and CAAT enhancer binding protein (C/EBP) were the most enriched (Fig. 8c). Interestingly, FOXA1 and FOXA2 are pioneer TFs that possess the ability to bind DNA sequences wrapped around nucleosomes and that are essential for the function of several cell types including liver cells[36,37]. They are enriched at enhancer and TSS and cooperate with master TFs to recruit Mediator and SMC1A at actively transcribed genes[38]. E1A induces high rate of

early viral promoters transcription through its interaction with Mediator and the formation of the preinitiation complexes[39]. Whether factor such as Mediator or cohesin present at FOXA binding motifs influence contacts between Ad5 and cellular chromatin or participate in Ad5 transcriptional activation via E1A interaction or both will need further investigation. It is also interesting to note that the second most enriched motif, CEBP motif, also known as CAAT box, that presents in a large proportion of cellular enhancers and promoters also regulates the expression of Adenovirus major late promoter (MLP)[40]. Again whether CAAT binding factors are involved in Ad5/cellular DNA contacts or only modulate Ad5 transcriptional regulation will need further analysis.

**Impact of Ad5 DNA/ host genome contacts on the cellular metabolism**. To search whether viral/host genome contacts impact cellular gene expression, we investigated different sets of genes whose TSS were in contact with Ad5 (Fig. 9). TSS from highly expressed genes were enriched in contacts, as well as TSS of genes were upregulated in fibroblasts infected by Ad5 (Fig. 9). The upregulation of some of these genes (CCNE2, CCND2, CAMKK1, ABTB1, HERC5, and ZAR1) was confirmed in infected PHH (Supplementary Fig. 15). Whether the contacts with the Ad5 genome directly promote gene upregulation or Ad5 preferentially positions at genes that are upregulated in infected cells or both remain to be determined. But altogether, these findings show that Ad5 contacts preferentially transcriptionally active regions, potentially altering their expression.

## Discussion

Here we applied Hi-C on HBV- or Ad5-infected PHH cells to study the interactions between these viruses and cellular host genomes. We show that both viral DNA molecules do not randomly position themselves in the nucleus, but instead establish preferential contacts with active chromatin regions, with the HBV genome presenting more specific contacts with CpG-rich regions. Intra- and inter-chromosomal spatial clustering of CGIs interphase cells has been suggested from 4C-seq experiments[41]. Since the HBV genome carries three CGIs, it is possible that these contribute to the recruitment of cccDNAs molecules to the host genome CGIs and active chromatin regions through the sharing of TFs, as was suggested for regions containing binding sites for CTCF[41], KLF1[42], or NF-kB[43]. Similarly to HBV, the Ad5 genome

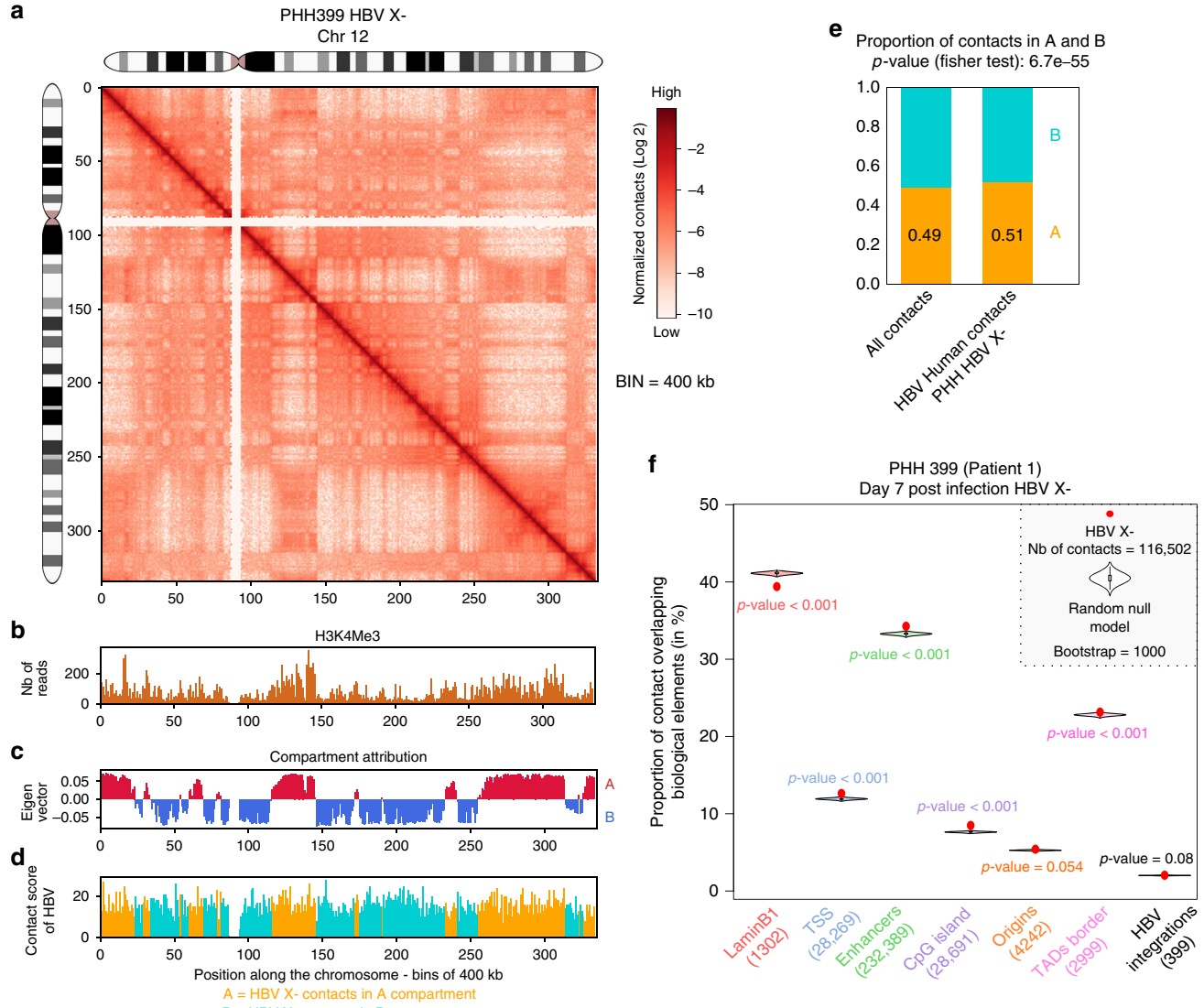

**Fig. 6** HBV X- genome contacts preferentially active chromatin at CpG-rich regions. **a** Hi-C contact maps of chromosome 12 from PHH infected with HBV X- at 400 kb resolution. The color scale represents the normalized contact frequencies between bins. **b** Active histone mark (H3K4Me3) reads density along chromosome 12 in HBV-infected PHH from Tropberger et al.[13] (400 kb resolution). **c** First principal component of chromosome 12 (400 kb resolution) representing the active A-type (red) and repressed B-type compartmentalization (blue). **d** Distribution of HBV contacts along chromosome 12 in infected PHH (right). Contacts with active chromatin (**a**) are shown in orange and with repressed chromatin (B-type) in green (400 kb resolution). **e** Distribution of Hi-C reads coverage in compartments A and B and distribution of HBV X- reads contacts on the genome in infected PHH (400 kb resolution). Statistical analysis was performed using Fisher test. **f** Proportion of HBV X contacts with a given biological element (as indicated along with the number of elements) compared with the corresponding null model that shows the distribution using the Hi-C coverage of the corresponding library and represented by violin plot. Density plot width = frequency, line = 95% confidence interval, box plot = interquartile range. Red dot represents the proportion of HBV X- contacts that overlap with the indicated biological element (computed in windows ± 3.5 kb around the start of the read). p-values were determined according to the percentages detected in the bootstrap strategy (see Methods)

displays preferential contacts with active chromatin, but repositions preferentially at enhancers and TSSs. We observed that contacted regions are enriched for FOXA factors known to recruit Mediator and cohesin[38]. Whether FOXA and its associated factors influence Ad5 targeting remains to be determined. It has been suggested that genes clustering may be established by chromatin bridging proteins, such as cohesin[44].

Alternatively, viral proteins could be involved in the positioning of viral DNA at active chromatin regions as was shown for the human immunodeficiency type 1 (HIV-1) integrase that forms a complex with the cellular factor lens epithelium-derived growth factor (LEDGF)/p75 to promote the integration of HIV-1

into active gene-dense chromatin regions[45]. HBV cccDNA is bound by viral proteins such as the regulatory protein HBx and the capsid protein HBc that have both been shown to be recruited onto cellular promoters[10,28,31,46,47]. We showed here that HBV X- positions at CGIs similarly to HBV wild type, showing that HBx is not involved in the contacts between the viral DNA and the cellular genome. Surprisingly, while HBV X- contains repressive chromatin marks[10], it is found associated with active chromatin. However, Lucifora and collaborators have shown that transcriptional repression of the HBV X- cccDNA is established only a few days after infection, suggesting that cccDNA may be first accessible to cellular factors or viral factors allowing its

targeting at active chromatin[27]. Whether other viral proteins such as HBc play a role in the targeting of HBV at active chromatin regions awaits further investigation.

HBV and Ad5 position at different active chromatin regions, suggesting that they provide a suitable and virus-specific environment for their transcription/replication. CGIs are bound by Cfp1 factors that maintain the region into a transcriptionally active state[22,23]. As a consequence, we propose that the preferential repositioning of cccDNA to CpG-rich regions favors the recruitment of Cfp1 to the viral molecule to establish transcription. Cfp1 contributes to transcription activation in part through the recruitment of Set1, which deposits H3K4me3[22,23]. We showed here that Cfp1 depletion in the host cell correlates with both the decrease of HBV transcription and H3K4me3 deposition

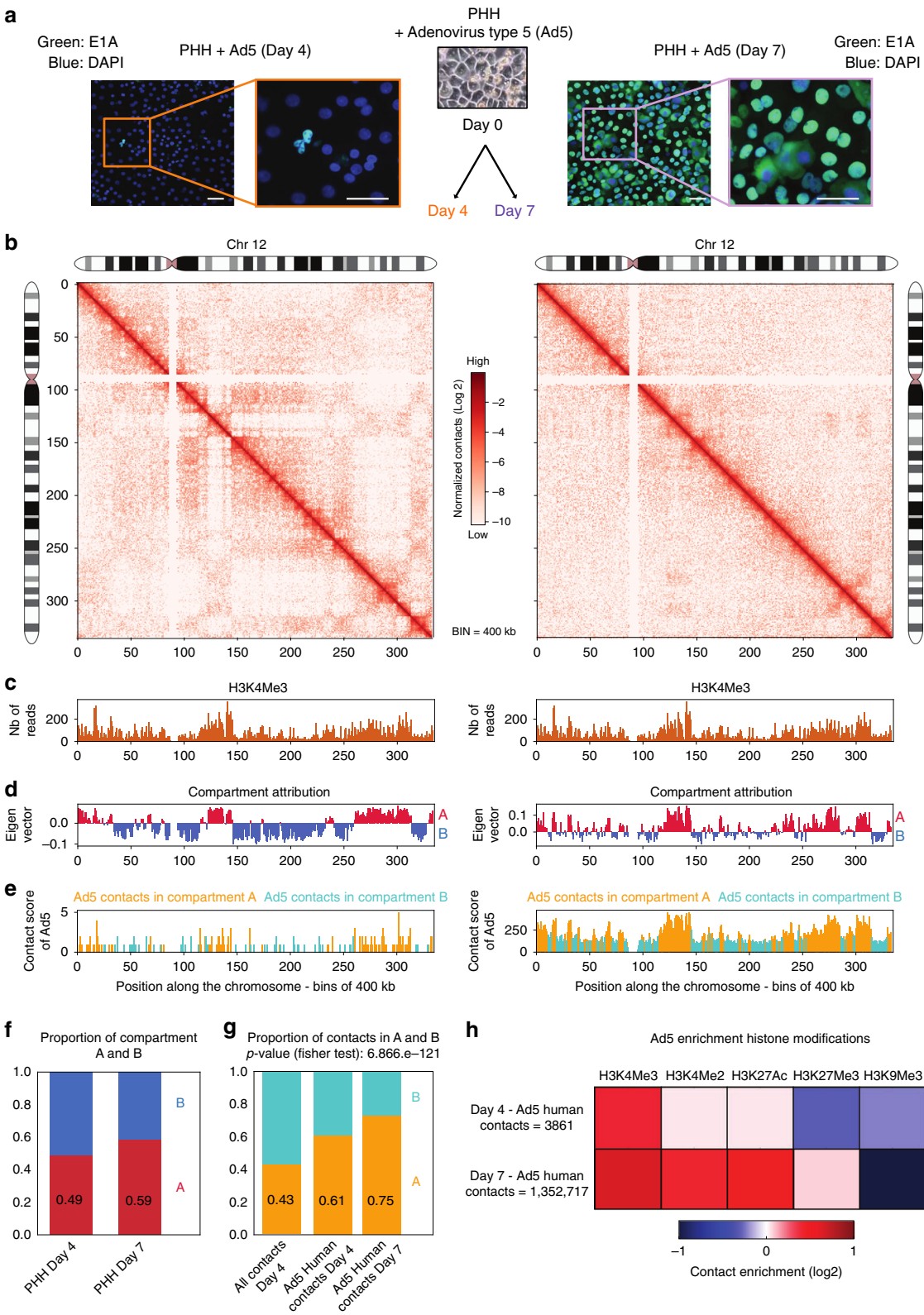

**Fig. 7** Distribution of Ad5 contacts along the host genome. **a** E1A protein (green) was visualized using immunofluorescence in PHH 345 at day 4 and day 7 p.p. Scale bar represents 50 μm. **b** Hi-C contact maps of chromosome 12 from PHH infected by Ad5 at 4 and 7 days after plating at 400 kb resolution. The color scale represents the normalized interaction frequencies between bins. **c** H3K4me3 reads density along the chromosome 12 using dataset from the Encode project generated using hepatic HepG2 cells (400 kb resolution). **d** First principal component of chromosome 12 (400 kb resolution) representing the active A-type (red) and repressed B-type compartmentalization. **e** Distribution of Ad5 contacts along chromosome 12 (400 kb resolution) in PHH replication Ad5 4 days (left panel) and 7 days after plating (right panel). **f** Distribution of active A and inactive B compartments in PHH at 4 and 7 days after plating (400 kb resolution). **g** Ad5 contacts are enriched in active compartment. Distribution of Hi-C coverage (400 kb resolution) in compartments A and B in PHH at day 4 post plating and distribution of Ad5 contacts on all genomes in PHH 4 and 7 days after plating. *p*-value was determined using Fisher's test. **h** Heatmap of Ad5 contacts enrichment on different histone modification marks (computed in windows ± 3.5 kb around the start of the read; *N* realizations = 100). Signals for the heatmap are represented by fold change (log2 ratio) compared with the average of random groups of contacts. Histone modification marks data were from the Encode project generated using hepatic HepG2 cells (Encode Project:GSE29611)[15]

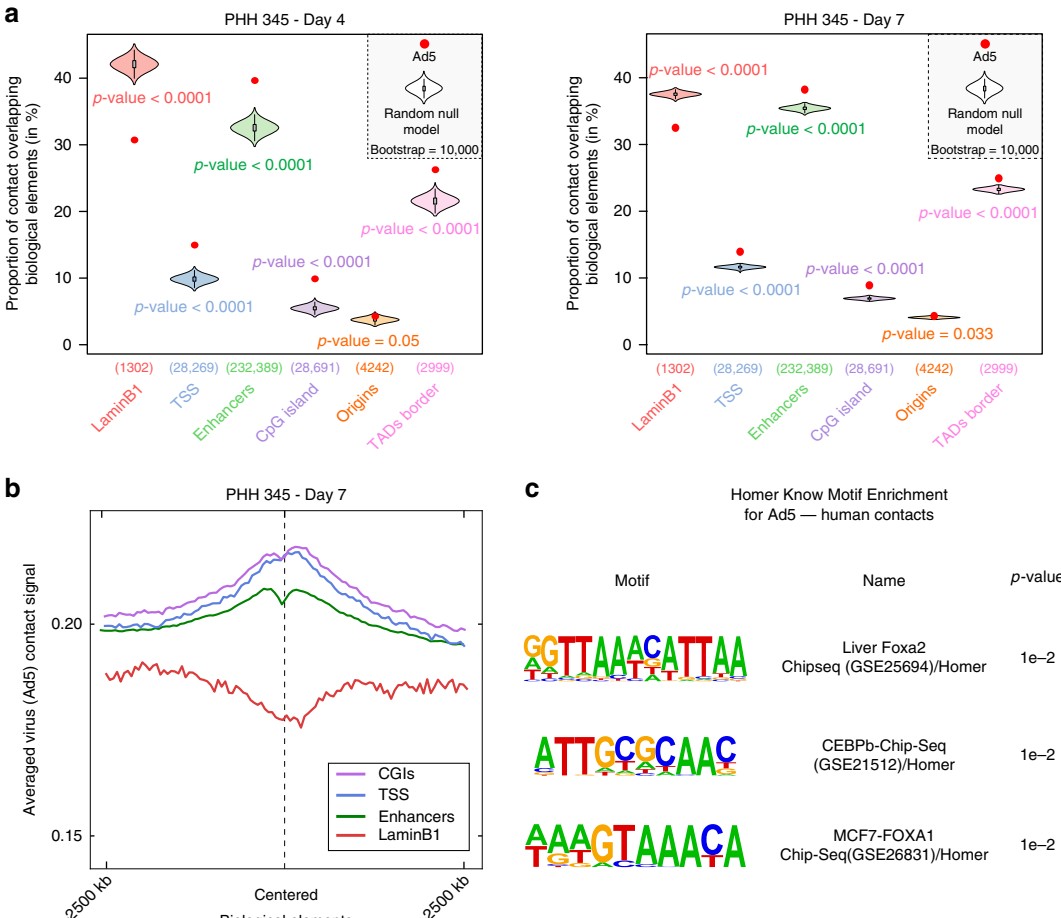

**Fig. 8** Episomal Ad5 DNA is enriched at enhancers and TSS. **a** Proportion of Ad5 contacts with a given biological element (as indicated on the graph along with the number of elements) in PHH at day 4 (all contacts) or day 7 (the number of contacts has been sub-sampled to 30,000) after plating compared with the corresponding null model that shows the distribution using the Hi-C coverage and represented by violin plot. Density plot width = frequency, line = 95% confidence interval, box plot = interquartile range. Red dot represents the proportion of Ad5 contacts with the indicated biological element (computed in windows ± 3.5 kb around the start of the read). *p*-values were determined according to the percentages detected in the bootstrap strategy (see Methods). **b** Average density plot of Ad5 contacts in PHH harvested 7 days after plating centered on different biological elements (50 kb resolution) of the host genome as indicated in the figure. **c** Enrichment of transcription factor (TF) motive identified by HOMER for Ad5 human contacts at Day 7 in PHH345

on the cccDNA, suggesting that Cfp1 is necessary for the recruitment of Set1 to the viral DNA so it can be properly transcribed as has been already demonstrated for cellular CGIs[23]. This is further supported by data showing that cccDNA transcription is controlled by Set1 recruitment[24]. As for the Ad5 genome, it preferentially repositions at TSS and enhancers, suggesting that this virus may also exploits some local molecular TFs to promote its own transcription. We observed that Ad5 contacts

regions enriched for FOXA and CEBP binding motifs. While we do not know whether FOXA controls Ad5 transcription, it has been shown that it recruits factors required for transcriptional activation by E1A[39]. It will be interesting to further analyze whether FOXA and interacting partners modulate Ad5 transcription in PHH.

Viral–host genomes interactions in HBV-infected PHH may also influence the expression of cellular genes, as suggested by the

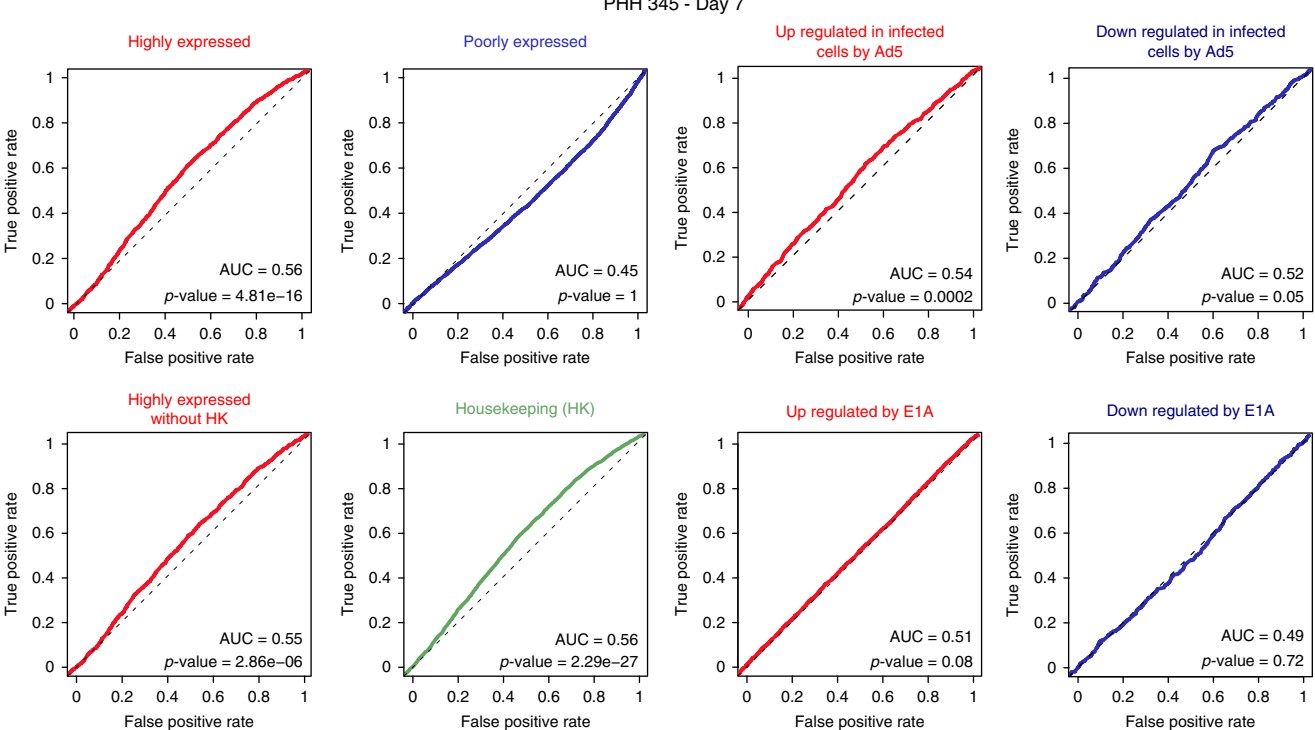

**Fig. 9** Ad5 contacts are enriched at TSS associated with highly expressed genes and upregulated genes. To measure enrichment of TSS highly contacted by Ad5 with respect to a given class of genes (as indicated on the graphs), we used receiver operating curve (ROC). The *p*-values of ROC analysis were computed with the Wilcoxon–Mann–Whitney test

enrichment of CGIs associated with deregulated genes in the set of HBV contacts. Similarly, the set of genes whose TSS are enriched in Ad5 contacts are upregulated in Ad5 infected arrested human fibroblasts, suggesting a causal relationship between viral contacts and gene expression changes. This set of observation raise the possibility that viral genomes interfere with cellular genes expression via DNA/DNA contacts. It will be interesting next to study how viral DNA contacts modulate cellular gene expression. Several studies identified clustering of co-regulated genes that cooperate with each other to enhance gene expression[48,49]. Moreover it has been shown that some genes in the cluster can influence the transcription of the other, highlighting the importance of this 3D organization in transcriptional regulation[49,50]. The mechanism of cooperation is not yet well understood but it is believed that the clustering may generate transcriptional areas with high local concentration of factors regulating transcription and creating thus a specialized micro-environment allowing efficient transcription. These mechanisms have been observed in the context of transcriptional activation of interferon type 1 beta (IFN-β). Virus infection triggers a NF-κB-dependent association between the INF-β gene and loci containing NF-kB binding sites and subsequent transcriptional activation[43]. DNA viruses may enhance the formation of such "transcriptional areas" by favoring the recruitment of cellular factors. In the case of HBV, we observed that HBV contacts are enriched at both CGIs associated with genes that are upregulated, but also downregulated upon HBV infection. It will be interesting to analyze whether HBV DNA could disrupt the spatial clustering of CGIs[41] leading to the downregulation of some cellular genes.

Overall, this work provides a global overview of the physical contacts between two viral genomes and their host, as well as insights regarding the functional significance of these interactions. We propose a model where DNA viruses use different strategies to infiltrate the genomic 3D networks and contact active chromatin. We showed that HBV cccDNA contacts CpG

islands known to be enriched for Cfp1, a factor involved in HBV transcription. This observation suggests that these viruses target specific active chromatin regions because these regions may provide an environment propitious to their own transcription/replication (Fig. 10). This hypothesis will be addressed in future studies, along with the mechanisms driving the repositioning of viral DNA at discrete locations, as well as the functional significance of the changes in gene expression observed in the host cell.

## Methods

**Cell culture, HBV production, and infection**. The HepAD38 cell line derives from HepG2 cells and contains the HBV genome (sub-type ayw) under tetracycline control[51]. HepAD38 cells were maintained in Dulbecco's modified Eagle medium/F-12 with 10% fetal calf serum (FCS), $3.5 \times 10^{-7}$ M hydrocortisone hemisuccinate, and insulin at 5 g per ml. Primary human hepatocytes (PHHs) were purchased from Corning (reference 454541, lot #399, #342, and #345) and maintained in the PHH medium (Corning, reference 355056, hepatocyte culture media kit, 500 ml) according to the manufacturer recommendations. HepG2-NTCP cells (A3 clone) derive from HepG2 cells and express the human sodium taurocholate cotransporting polypeptide (NTCP). HepG2-NTCP cells are grown in DMEM with 10% fetal calf serum (FCS)[52].

For virus production, HepAD38 cells were grown in Williams E medium supplemented with 5% FCS, $7 \times 10^{-5}$ M hydrocortisone hemisuccinate, 5 mg/ml insulin, and 2% dimethylsulfoxide. HBV particles were concentrated from the clarified supernatant through overnight precipitation in 5% PEG 8000, followed by centrifugation at 4 °C (60 min at 5292 × *g*). Titers of enveloped DNA-containing viral particles were determined by immunoprecipitation with an anti-preS1 antibody (gift of C. Sureau, dilution 1/2000), followed by qPCR quantification of viral RC-DNA using RC primers (Supplementary Table 3). For infection, only enveloped DNA-containing viral particles (vp) were taken into account to determine the multiplicity of infection (MOI). PHHs were infected for 7 days with normalized amounts of virus at an MOI of 500 vp/cell, as described[10]. To avoid neo-synthetized RC-DNA spreading and recycling, PHH were treated with 25 μM lamivudine (Sigma-Aldrich) for 3 days before processing the cells[53]. HepG2-NTCP cells were infected with normalized amounts of virus at an MOI of 100 vp/cell in presence of 4% PEG 8000. Twenty-four hours after inoculation, cells were washed with PBS 1X and transfected with the indicated siRNA as described in the siRNA section. Eight days after infection, HBV transcription was analyzed by RT-qPCR. The level of HBV RNA in HepG2-NTCP cells transfected with Si Ctr set to 1.

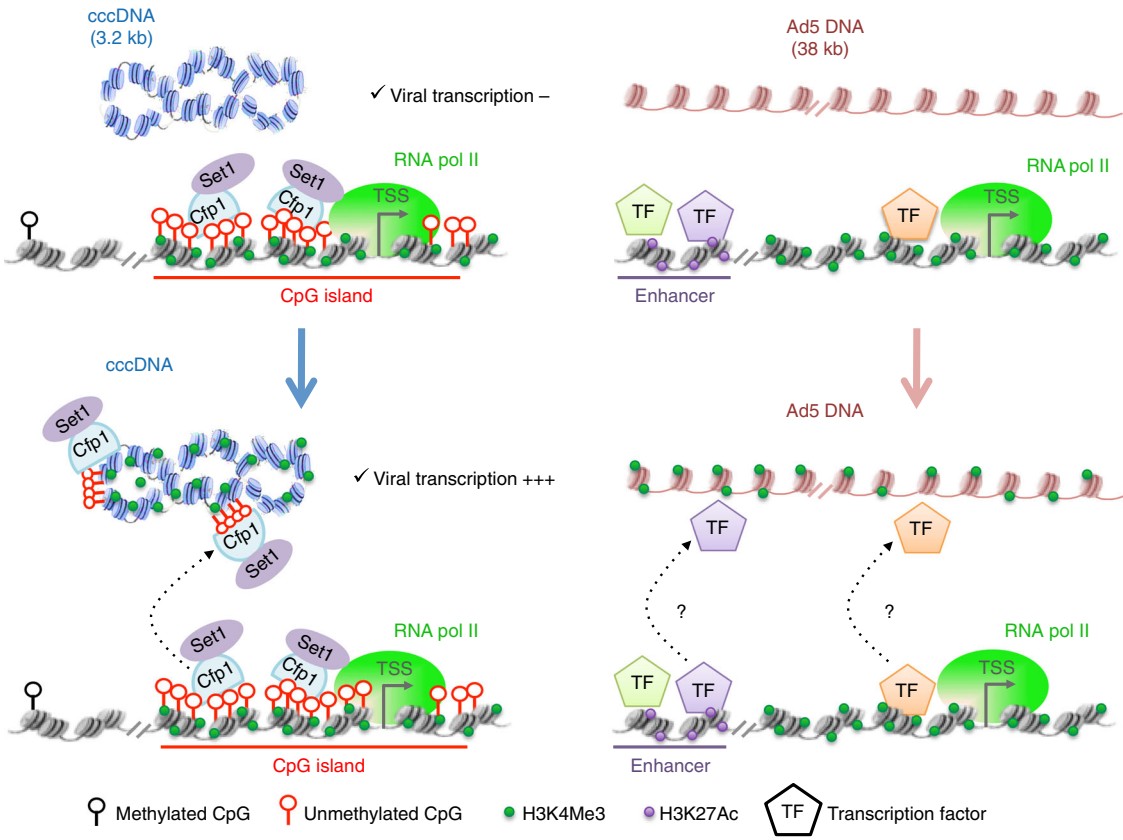

**Fig. 10** A proposed model showing that DNA viruses infiltrate the 3D network and contact active chromatin. Our model suggests that viruses target critical regions to favor their own replication/transcription

PHH (345) were seeded and maintained in the culture for 4 or 7 days. Ad5 infection was followed by measuring adenovirus gene expression of E1A and E1B (early phase) and protein II (also called hexon; late phase) (Methods), as well as by tracking the amount of DNA. E1A expression was also confirmed by immunostaining and immunoblotting.

**Quantitative RT-PCR (RT-qPCR).** Total RNA was prepared using TRIzol reagent (Invitrogen) and TURBO DNase (Ambion). RNA (500 ng) was retrotranscribed using random primers and RevertAid H Minus M-MuLV reverse transcriptase (Fermentas). RT-qPCR experiments were carried out as described[54]. For relative quantifications, Rhot2 was used as a reference gene because of its low variation coefficient in human liver tumors and cell lines[55]. Values were calculated according to the ΔCt quantification method with ΔCt = Ct HBV – Ct Rhot2. Results are expressed as the average of at least three independent experiments. Standard error of the mean (SEM) are indicated. p-values were determined by Mann–Whitney test. The primers HBV RNAall-F and HBV RNAall-R amplify all HBV transcripts (pgRNA as well as the 2.4 and 2.1 kb mRNA), except the 0.8 Kb transcript encoding HBx (Supplementary Table 3). Primers "pgRNA F" and "Pg RNA R" are specific for pgRNA (Supplementary Table 3). Primers for E1A and E1B RNA amplification are described (Supplementary Table 3).

**Quantification of cccDNA.** For nuclei isolation, cells were incubated for 15 min at 4 °C in buffer I (10 mM Hepes pH 8, 1.5 mM MgCl$_2$, 10 mM KCl), lysed by adding 1% NP-40, and shaken using a vortex for 30 s. Nuclei were pelleted by centrifugation (10 min at 1600 × g) and washed in buffer I. Following centrifugation, DNA was extracted using QIAamp DNA blood mini kit (Qiagen). For HBV cccDNA quantification, DNA was pre-treated with 10 U plasmid-safe DNase (Epicenter) for 1 h at 37 °C, and cccDNA was amplified using cccDNA primers (Supplementary Table 3).

**Chromatin immunoprecipitation (ChIP).** ChIP experiments were carried out on infected HepG2-NTCP 8 days post infection with the ChIP-IT High sensitivity kit (Active motif, 53040) with minor modifications. Cells were fixed with 1% formaldehyde for 15 min at room temperature (RT) and quenched with 0.125 M glycine for 5 min. For nuclear extracts preparation, cells were lysed in PBS supplemented with protease inhibitors and homogenized with CK14 Precellys beads (KT03961-1-203.05) at 6700 rpm, pulse for 20″ on and 30″ off, for 3 cycles. After centrifugation, nuclei were suspended in ChIP buffer (1% SDS, 10 mM EDTA,

50 mM Tris pH 8, supplemented with protease inhibitors). After sonication and centrifugation, chromatin was immunoprecipitated overnight at 4 °C using 4 μg of H3K4ME3 (miilipore 07-043) and anti-Cfp1 (ab 56035). Immunoprecipitations with irrelevant non-specific immunoglobulins (Millipore PP64B) were included in each experiment as a negative control. Immune complexes were then incubated with 30 μl of protein G agarose beads for 3 h at 4 °C. Immunoprecipitated complexes were washed and eluted in Elution buffer AM4. After purification of DNA, qPCR was performed using specific cccDNA primers (Supplementary Table 3). Samples were normalized to input DNA using the ΔCt method, where ΔCt = Ct (input) Ct (immunoprecipitation) and computed as a percentage of the input. Results are expressed as the average of at least three independent experiments. Standard error of the mean (SEM) are indicated. Statistical differences were analyzed by Mann–Whitney test.

**Immunofluorescence.** PHH were fixed with 4% paraformaldehyde for 15 min at RT and permeabilized and blocked with [3% BSA, 0.5% Triton X-100, 10% FBS in 1× PBS] for 30 min. Cells were washed 3 times with 0.1% Tween 20 in PBS and incubated for 2 h at RT with the primary antibody. After three washes in PBS 0.1% Tween 20, cells were incubated with a secondary antibody coupled to Alexa 488 for 45 min at RT. Coverslips were mounted with Vectashield (Vector Laboratories) supplemented with diamidino phenyl indol (DAPI) for nuclear staining. Fluorescent images were acquired on an Axio Observer.Z1 microscope with an Apotome Camera with Plan-Apochromate 63x/N.A. 1.40. Images were acquired with AxioVision software (Carl Zeiss, Germany).

**Hi-C library preparation.** Hi-C was carried out as described previously[12] with minor modifications. Five million PHH cells were collected and incubated for 10 min at RT in culture medium containing 1% formaldehyde. The reaction was stopped by addition of 2.5 M glycine (final concentration: 0.2 M) and incubated for 5 min at RT followed by 15 min on ice. The suspension of cross-linked cell was centrifuged at 300 × g for 10 min, and the pellet suspended in 500 μl ice-cold Hi-C lysis buffer (10 mM Tris-HCl pH 8.0, 10 mM NaCl, 0.2% Igepal CA630) with 100 μl of protease inhibitors (Sigma, P8340). The suspension was incubated on ice for 15 min, centrifuged (2500 × g for 5 min), and the cell pellet suspended in 1 ml of digestion buffer 1X (NEB). The cells were transferred in a 2 mL tube Soft tissue homogenizing CK14 (BERTIN Instruments), then lysed using tissue homogenizer (Precellys 24, BERTIN Instruments; 6700 rpm, 3 cycles 20 s on/30 s off, repeated twice). Lysate was transferred into loBind tubes (Eppendorf). Precellys beads were

washed with 800 μl NEB buffer subsequently pooled with lysate. Total lysate was split into six 300 μl aliquots. Each aliquot was mixed with 200 μl NEB buffer 1X, followed by the addition of 15 μl SDS 10% and incubation at 65 °C for 10 min. SDS was quenched by adding 100 μl of 10% Triton X-100 in each tubes which were kept at 37 °C for 30 min. Chromatin was digested overnight at 37 °C by adding 1000 Units of BglII (NEB). DNA ends were filed and labeled with biotin in klenow end-filling reaction with 1.5 μl of 10 mM dATP, 1.5 μl of 10 mM dGTP, 1.5 μl of 10 mM dTTP, 37.5 μl of 0.4 mM biotin-14-dCTP (Life Technologies, 19518018), and 40 U of Klenow (NEB, M0210) incubated at 37 °C for 45 min. Digested DNA was then ligated by adding 900 μl of ligation buffer containing 120 μl of 10X NEB T4 DNA ligase buffer, 100 μl of 10% Triton X-100, 12 μl of 10 mg per ml Bovine Serum Albumin, 663 μl of water, and 5 μl of 400 U per μl T4 DNA Ligase (NEB, M0202), and incubation at RT for 4 h under slow rotation. After cross-link reversal and protein degradation with proteinase K (65 °C, overnight), DNA was purified using phenol-chloroform extraction and suspended in TE 1X supplemented with RNase A (10 μg per mL final). DNA was then sheared using a Covaris LE220 apparatus (Duty cycle 15, intensity 150, Cycles/burst 200, time 80 s). After shearing DNA was run on 1% agarose gel and bands ranging from 300 to 800 pb were extracted from the gel using minielute gel extraction kit (Qiagen). DNA fragments were immobilized using MyOne Streptavidin C1 dynabeads (Life Technologies), end repaired, adenine tailed and ligated to custom made paired-end adapters (compatible with Illumina primers). The immobilized Hi-C DNA was amplified using Illumina primers PE PCR 1.0 and PE PCR 2.0 with 8–12 PCR amplification cycles. PCR products were further used for direct Illumina pair-end sequencing using nextSeq 500 High output kit v2 (Illumina), or processed through an additional step of capture C (CHi-C).

**Capture Hi-C protocol**. Capture Hi-C of HBV sequences were carried out using custom-designed xGen Lockdown probes (Integrated DNA Technology) of 120 bp, each covering the entire HBV genome genotype D with twofold coverage. A total of 500–1000 ng of HI-C DNA were enriched for HBV DNA sequences according to the manufacturers' xGen rapid capture protocol v2, using custom paired-end blockers. Fifty microliters of Dynabeads MyOne streptavidin C1 (Life Technologies) were used for capture reaction. DNA-enriched libraries were then amplified with four PCR cycles using PE PCR 1.0 and PE PCR 2.0 primers (KAPA HIFI HotStart ReadyMix) and sequenced using Illumina pair-end sequencing.

**Hi-C and capture Hi-C data analysis**. PCR duplicates for each Hi-C library were filtered using the random tags present on the custom oligonucleotides[56]. Read-pairs were aligned independently against the reference human genome (NCBI hg19) and the virus genomes using Bowtie[57], using an iterative procedure (very sensitive mode, mapping quality > 30). The HBV and adenovirus sequences used were HBVayw genotype D (https://www.ncbi.nlm.nih.gov/nuccore/V01460.1) and adenovirus C JX173077.1 Human adenovirus C strain human/ARG/A8649/2005/2 [P2H2F2], respectively.

To generate contact maps, each mapped position was assigned to a BglII restriction fragment (RF). Non-informative events such as self-circularized restriction fragments or uncut adjacents RFs were discarded by taking into account the read-pairs relative directions and the distribution of the different configurations[11,58]. The genome was binned into 100 kb or 400 kb segments, and the corresponding contact maps were generated and normalized using the sequential component normalization procedure (SCN)[11]. SCN alleviates the intrinsic experimental and sequencing variations of the Hi-C protocol (restriction sites distribution, repeated sequences, etc.) The normalized contact value between two bins corresponds to a number between 0 and 1 that corresponds to the relative contact frequency between this pair of bins compared with all other pairs involving one of those bins. The logarithmic color scale representation helps the visualization.

Eigen vector decomposition was done as follows: the effect of genomic distance was calculated with the normalized 400-kb binned matrix for each chromosome. Each element of the latter matrix was detrended by dividing it with the expected value according to the genomic distance effect. The correlation matrix was then generated and the eigen values and first eigen vector were computed. The latter vector allows the compartment A or B attribution as previously shown in[57]. Positive and negative eigen values were arbitrarily chosen to correspond to the A and B compartments, respectively. This was done using a positive Pearson correlation coefficient with a signal of histone marks associated with active chromatin (H3K4Me3).

For the computing and display of the contact signal profile of virus genome along human chromosomes, we assigned each pairs of read-pairs involving virus and human chromosome to the corresponding 400 kb bins.

For HBV profile, Hi-C and CHi-C data were merged.

**Histone mark enrichment plot**. Histone modification ChIP-seq datasets were recovered from publicly available databases, and the corresponding reads are aligned along the NCBI hg19 genome using Bowtie. For each contact made by the virus, the number of reads of the histone mark signal was aggregated over a 7-kb window centered on the position of contact (windows ± 3.5 kb around start of read). The same procedure was applied for the corresponding input signal. The ratio of both sums provides the strength of contacts between the virus and a given histone mark. This value was compared with values obtained using a null (random) model. As for the overlap strategy (below), the genomic positions of random positions were chosen by taking into account the differences in detectability of each Hi-C experiment (i.e., differences in local digestion, cross-link efficiencies). All the contacts detected in a same Hi-C library were loaded into computer memory. A group of positions was then randomly picked (with replacement) among them to have the same number of virus-human chromosomes contacts. Doing this, a group of fragments that has been detected a lot of times in a Hi-C library will also have more chances to be picked in the random positions generations respecting the difference of detectability between genomic regions inherent to a Hi-C experiment. The mean of 100 realizations was computed to estimate the expected strength of contact for genomic positions of random group. The log2 ratio between the virus contact strength and the random positions strength is computed and plotted for five histone marks.

The dataset used in this work correspond to the following references on the short-read archive (SRA) database: SRR227563 for H3K4me3, SRR227466 for H3K4me2, SRR227575 for H3K27ac, SRR227598 for H3K27me3, SRR568329 for H3K9me3, and SRR227552 for input.

**Co-localization analysis**. We developed a bootstrap strategy to determine whether the proportion of contacts made by the virus and overlapping a biological set of genomic positions was significant. The pair-end reads for which one read of the pair contacts the virus and the other read the human genome were processed as follow. First, 7 kb windows centered on the start of each read mapped on the human genome were selected and for each window, the number of positions of various categories of functional elements was computed (lamins, TSS, Enhancers, CpG, replication origins, TADs borders, and integration of HBV in non-tumoral cells). The positions of lamins (track: laminB1Lads), TSS (track: refGene_hg19_TSS, 7 kb window centered on the positions), CpG (track: cpgIslandExt), origins of replication (file used: wa.HepG2.rep-1.J7.hg19.pks.bed, 20 kb window centered on the position) were retrieved from USCS database on the hg19 assembly. Enhancers positions were extracted from the ENCODE project (https://www.encodeproject.org/files/ENCFF558TCP, file: hepatocytes-DS32057A.peaks.fdr0.01.hg19.bed. For a group of N contacts between virus and human chromosomes, we generate a group of N random contacts by sampling with replacement the group of positions already detected in the whole Hi-C library. Doing that, a genomic region that is well detectable (and that will have more detected positions in the whole Hi-C library) will have more chance to be picked up and in contact with the virus. We reiterate this procedure 1000 or 10,000 times. From the set of these realizations, we computed the expected distribution of proportion of contacts overlapping a biological group and displayed the result as a violin plot. A p-value was computed by comparing the detected proportion of contact involving the virus (HBV or Ad5).

TADs boundaries were determined with the HiCseg software[59] based on a block-wise segmentation model. The software was run on SCN normalized contact map of each chromosome (bin: 100 kb) with the following parameters: "P" for Poisson distribution, "Dplus" for the extended block-diagonal model and the maximal number of change-points was set to the size of the matrice divided by five.

The positions of HBV integration within the genome of non-tumoral cells were retrieved from the work of[16]. The duplicates in the biological group positions were removed using R "unique" function.

CGIs associated with highly and poorly expressed genes were established as follows: the list of transcript expressions was sorted according to their values in FPKM. Then, the 5000 first elements and the 5000 last elements were retained. Duplicated genes were removed. CGIs were associated to their corresponding genes with windows ± 5 kb surrounding the TSS which gives a total of 3698 CGI and 2213 CGI for highly expressed genes and poorly expressed genes, respectively. The group of deregulated genes includes genes whose log2 ratio of expression was above +2.0 (upregulated) or below −2.0 (downregulated). This procedure allows us to have large groups, which was necessary to have significance.

The number of overlaps with the set of positions of a group of genomic features was computed with the function "findOverlaps" from the R package GenomicRanges. The null models were built as followed: genomic positions of all the contacts from a Hi-C library were loaded into memory. From this set of genomic positions, a group of positions with the same number of virus contacts were randomly picked with replacement. This procedure allows to take into account variations in local detectability due to the Hi-C protocol (crosslink, digestion efficiencies, etc.) 10,000 or 1000 random realizations were generated and the p-value estimated from the number of realizations with a contact strength above or equal to the one detected with the virus contacts group.

**Enrichment visualization plot**. To visualize virus contact profile around positions of interest, we summed all the contacts made by the virus with 50 kb bins within 5 Mb windows centered on the positions of genomic features of interest (TSS, CpG, origin of replication, enhancer, and LaminsB1). To normalize for variability in detectability, we divide the previous signal by the Hi-C coverage of the same bins.

**Functional annotation enrichment analysis of CGI in contacts with the HBV genome**. TSS/CGI genes in contact with the viral genome were analyzed using the Database for Annotation, Visualization and Integrated Discovery (DAVID) tool (v6.8)[60]. Gene enrichment was evaluated by using Agilent CpG island gene list provided by the database as background.

**Motif enrichment analysis**. To identify recognition binding sequences of known transcription factors in regions enriched in contacts with the virus, we binned the contact signal of Ad5 at 2 kb resolution. We computed the Hi-C coverage at the same resolution. Peaks of enriched contacts for Ad5 were defined as the ones with a ratio of contact signal over the Hi-C coverage above five, resulting in 2976 sequences. The genomic positions of the corresponding bins along the hg19 genome were used as input for the HOMER software[61] with default parameters (hg19 ER_MotifOutput/ -size 200 -mask).

**ROC analysis**. We applied a receiver operating characteristic (ROC) analysis to test for significant associations between TSSs enriched in contacts with Ad5 and genomic features of interest (highly expressed, poorly expressed, housekeeping, deregulated, upregulated, and downregulated genes in Ad5 infection). TSSs were sorted according to a "virus occupancy score" computed by summing the number of contacts done by the virus around a TSS (centered window of 7 kb) divided by the Hi-C coverage of the region (number of all Hi-C contacts in the same window). From this ranked list, each TSS is labeled "true positive rate" if TSS belongs to a type of genomic features, and "false positive rate" if not. The proportion of positive and negative TSS is plotted along the ranked list of virus occupancy scores.

Transcriptomics data from GSE32340 study (GSE32340_RPKM_mock_e1a_samples.txt) were used to determine highly and poorly expressed as well as up- and downregulated genes with E1A protein infection.

RNAseq analysis of PHH was used to group genes according to their expression levels, with 3249 highly expressed genes corresponding to genes whose expression is above 20 RPKM (Reads Per Kilobase per Million mapped reads). 20,141 poorly expressed genes were pooled together, corresponding to genes whose expression levels are below 0.5 RPKM.

Genes up- and downregulated during Ad5 infection were defined as those displaying a log2 fold change superior to +1 or inferior to −1, respectively, in the transcriptomics data[32]. Genes up- and downregulated with E1A protein infection were defined as those displaying a log2 fold change superior to +1 or inferior to −1, respectively. Housekeeping genes were retrieved from[62].

**siRNA transfections and siRNA**. siRNAs control (ON-TARGETplus Non-targeting Pool #D-001810-10-0X) and siRNA directed against cfp1 were purchased from Santa Cruz Biotechnology. Twenty-four hours after infection, HepG2 NTCP cells were transfected with 40 nM of si Ctr or si Cfp1 (sc-35055) using Lipofectamine RNAiMAX (Invitrogen; 17-0618-0) reagent according to the manufacturer's protocol. RNA was extracted 8 days after infection.

**RNAseq analysis**. Three biological replicates were used for each conditions (infected and non-infected). PHH were infected as described above. Total RNA was extracted and purified using TRIzol reagent (Invitrogen) and TURBO DNase (Ambion). cDNA library, sequencing using illumina PE 150 and RNAseq analysis were performed by Novogen. On average, ~95 million reads were generated per sample. Clean reads were aligned to the reference genome with Tophat2[63]. The expression of lncRNAs and mRNA was assessed by Cuffdiff (http://cole-trapnell-lab.github.io/cufflinks/cuffdiff/index.html). Square of Pearson Correlation Coefficient between the biological replicates was >0.94. Differentially expressed genes had a $q$ value < 0.05. Enrichment pathways were analyzed using KEGG enrichment analysis.

**Antibodies and reagent**. Anti-tubulin (Dilution 1/10,000 for western blot WB) was purchased from SIGMA (Catalog number T5168), Anti-beta actin (Dilution 1/10,000 for western blot WB) was purchased from SIGMA (Catalog number A5316), anti-E1A (dilution 1/1000 for WB and 1/200 for immunofluorescence IF) from Santa Cruz Biotechnology (Catalog number sc-58658, lot # G2611), anti-CGBP (Cfp1; dilution 1/1000 for WB) from Santa Cruz Biotechnology (Catalog number sc-136419, lot # H0510), HBc (Dilution 1/30,000 for WB and 1/500 for IF) was from DAKO (Catalog number B0586, lot # 1018412) and prediluted anti-HBs (no dilution) from sigma (Catalog number ab859, lot # GR3199714-3).

**Western blot**. Samples were resolved by SDS-PAGE and electro-transferred to nitrocellulose membranes. Blots were incubated with the indicated primary antibodies and then probed with Dye-conjugated secondary antibodies. Fluorescent immunoblot images were acquired and quantified by using an Odyssey scanner and Odyssey 3.1 software (Li-CorBiosciences). Uncropped scans of blots are reported in the Supplementary Information as Supplementary Figure 16.

**ChIP-seq datasets used**. Histone modification marks dataset were from the Encode project generated using hepatic HepG2 cells (GSE29611).

**Code availability**. Codes produced in this study can be accessed at https://github.com/axelcournac/virus_Hi-C_Analysis.

## Data availability

Sequencing data for the RNA-seq and Hi-C libraries have been deposited in Short Read Archive under project number PRJNA488420. The authors declare that all other data supporting the findings of this study are available within the article and its Supplementary Information files, or are available from the authors upon request.

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

## Acknowledgements

We thank C. Seeger and S. Urban for kindly providing cell lines used in this study. We thank the UTechS PBI (Imagopole)-DTPS/C2RT, part of the France-BioImaging infrastructure network supported by the ANR (ANR-10-INSB-04; Investments for theFuture), for the use of the Zeiss widefield apotome microscope. This work was supported by funding to C.N. from the Agence Nationale de la Recherche sur le Sida et les Hépatites Virales (ANRS) and Fondation pour la Recherche sur le Cancer (ARC). This research was also supported by funding to R.K. from the European Research Council under the 7th and H2020 Framework Program (FP7/2007-2013, ERC grant agreement 260822 and H2020 ERC grant agreement DLV-771813), and from Agence Nationale pour la Recherche (MeioRec ANR-13-BSV6-0012). P.M. was supported by ANRS and Institut Carnot. This study makes use of data generated by the ENCODE Consortium and the ENCODE production laboratories.

## Author contributions

P.M. and A.C. performed the experiments and analyzed data, with contributions and help from G.A.P., A.T., S.C. and S.M., M.L. and M.M. contributed to experimental design. R.K. and C.N. designed experiments, analyzed data and wrote the manuscript with P.M. and A.C.

## Additional information

**Competing interests:** The authors declare no competing interests.

