## [Peer Review File · Nature Communications]

Reviewers' comments:

Reviewer #1 (Remarks to the Author):

This manuscript describes DNA-DNA contacts between non-integrating viral genomes and human host cell DNA using high-throughput chromosome conformation capture technologies such as Hi-C and capture Hi-C. The authors investigate primary human hepatocytes infected with either of two viruses, HBV or Adenovirus at different times post-infection. They report that overall host cell TAD structure remains largely similar before and after infection but the viral genomes show a non-random interaction distribution across the host genome. The viral genomes tend to interact with open chromatin regions, enriching in compartment A contacts, more so for Adenovirus than HBV. But the two viruses show preferential associations with different parts of the genome. The HBV genome interacts with CpG islands whereas Adenovirus genome interacts with transcription start sites and enhancer regions in addition to CpG islands. The authors further show that such DNA-DNA contacts are associated with highly expressed genes. The authors propose that the purpose of such contacts may be to facilitate transfer of transcription factors and other gene regulatory factors from the host genome to viral DNA.

The Hi-C data have good quality and are analyzed properly. The non-random DNA-DNA contacts are interesting and hint at potentially significant but unknown mechanisms for viral DNA targeting to the human genome. However, all presented data are correlative. No evidence is presented to support the model in figure 5. I have no problem with a speculative model but as presented, the authors give the impression that the data support their model. This is not the case. They have not shown that impeding viral-host genome contacts would adversely affect viral genome replication. They show that Cfp1 is required for effective HBV genome production but it is not clear if the DNA-DNA contacts play a role here.

Other issues to consider:

Why are there many more interactions between the Ad5 and host genomes than HBV-host? Are there differences in MOI or number of infecting particles? If so, would HBV show similar contact distribution if it was present at high copy number?

The authors indicate that "HBV contacts interfere with gene expression" in the abstract. This is just a conjecture and should be included only as part of the discussion. The same is true for Ad5.

On page 3, the authors indicate that Cfp1 regulates HBV transcription through H3K4me3. Again, the data is correlative and thus the statement is technically not correct.

There are numerous typos in the manuscript.

Please include lot numbers for the antibodies used in the paper in addition to the catalogue numbers.

Reviewer #2 (Remarks to the Author):

The authors analyzed the chromatin interactions in primary human hepatocytes (PHH) infected by either hepatitis B virus (HBV) or adenovirus type 5 (Ad5) virus. They showed the different chromatin positioning of the two viruses to the host genome. The results demonstrated how DNA viruses use different strategies to target the host genomic regions for favoring their growth. Understanding the mechanisms of viral DNA integrated to the host genome is an important problem. This study provides a new aspect from the 3D genome to reveal the interaction between the viral DNA and the host genome upon viral infection and how the host transcriptome is hijacked by the virus. These observations are intriguing and the data are interesting.

1. The major concern of the analysis is the resolution of the Hi-C and CHi-C. The resolution of the Hi-C is 400kbp, which is pretty low. The resolution of the CHi-C cannot be found in the manuscript or the supplementary materials. All the figures show the analysis results using 400kbp bins. At this resolution, it is unclear how to pinpoint the viral-host contacts. Within 400kbp genomic range, there can exist diverse functional regions including CpG islands, promoters and enhancers. Or the viral DNAs are dozens of kbp, which allows pinpointing the viral-host interaction at a much higher resolution? Since this is the foundation of all the analyses, the details of determining viral-host contacts need to be provided.

2. Processing Hi-C and CHi-C data can be tricky. The supplementary method does not provide any critical detail such as normalization, correction of sequencing bias, call of contacts etc. It only mentioned to use a published SCN method to process the Hi-C data. How is this program compared to the other Hi-C processing packages? What are the parameters used for processing this data set?

3. The change of TAD boundaries can be a very interesting observation after Ad5 infection. But the TAD boundary is around several kbp long. Given the 400kbp resolution of the host Hi-C data, how to determine the change of TAD boundary?

Minor points:

1. The sequencing reads and depth of Hi-C and CHi-C are not described in Supplement.

Reviewer #3 (Remarks to the Author):

The manuscript entitled "DNA viruses infiltrate the host genomic 3D network to shape the host transcriptome and to favor their replication" by the authors Moreau et al. focuses in investigate whether DNA viruses that do not integrate into the host genome take advantage of interacting with specific sites of the host genome. The authors used two DNA viruses as model: hepatitis B virus (HBV) and Adenovirus serotype 5 (Ad5) and characterized their

contact sites with the host genome. By using the Hi-C technique and viral DNA-capture, the authors concluded that HBV preferentially contacts CpG islands, whereas Ad5 transcription start sites and enhancers. The manuscript is an elegant characterization of the sites where HBV and Ad5 genomes interact with the host genome. However, the work is descriptive and, before it can be considered for publication in Nature Communications, it would require mechanistic analysis. For example, what is the consequence of the described interaction?, how is this interaction facilitated?, are there viral factors or cellular factors involved?, the interactions observed are cause or consequences? The title does not reflect the findings of the manuscript: there is no indication that the interaction between viral and host genome affects both the host transcriptome and viral replication. It is not clear whether the interaction between viral and host genome affects the host transcription or whether this is a result of the viral expression. At the moment the manuscript only shows correlations.

In addition, the following aspects should be considered:

1. References 2 and 3 in the abstract are incorrect.
2. Figure 1a and Supplementary Figure 1 show HBs staining in the uninfected PHH cells. At least two explanations can be proposed for this observation, the PHH cells derive from a chronically HBV infected patient, or, the HBs antibody is not-specific. This should be discussed.
3. At the end of page 2, the authors mentioned that "At this resolution, the organization of the genome of hepatocytes appears similar to those of other cell types, whether the cells are infected or not". However, the authors did not compare cell types, instead two different PHH are compared. Therefore, this needs to be corrected.
4. Figure 1F: please, explain the meaning of the orange and green colors.
5. What percent of the total HBV interactions with host genome corresponds to CpG islands? This data will help to get an idea of the preference for the binding to CGI sites with respect to the whole interactions.
6. Figure 2C right (pathway enrichment of deregulated genes), the panel is not mentioned in the text.
7. Figure 2D shows total HBV contacts in the context of biological elements (example: upregulated, and downregulated genes). It is also important to show, out of the 1139 mRNAs and 31 LncRNAs affected RNAs, what % of those genes actually contacts HBV.
8. Figure 2E, western blots of siControl and siCfp1 (right panel) are not correctly labeled. In addition, the left panel, "ttx/roth" and "pg/roth" needs to be explained.
9. Several groups, including this group, have shown that the viral protein HBx is important in regulating the chromatin state of cccDNA. Thus, how do HBx and Cfp1 relate in the HBV context? What would happen if the infection is performed with an HBV mutant defective for HBx expression (HBV X-)? Do the HBV and host genome interactions change under conditions where HBx is not present? Does HBx play any role in recruiting HBV to CGIs regions? Does HBx interact with Cfp1? These questions could help in clarifying some of the mechanistic aspects missing in this manuscript.
10. It is not clear to me what the authors want to say with "PHH donor (345) display a natural infection by the human adenovirus serotype 5...". The other PHH cells cannot be infected with Ad5? Please, clarify this.
11. There is no description regarding the PHH infection with Ad5 as well as how the Ad5 virus was produced.

12. The presentation of the Ad5 data is confusing, going back and forward from Figure 3 to Figure 4. This should be corrected.
13. Figure 3C is not discussed in the text.
14. In page 5, second paragraph, it is mentioned that "...Ad5 contacted preferentially regions enriched for active marks at promoters and enhancers..." and refers to Figure 3h. However, this figure only shows enrichment on histone modifications globally, without looking at enhancers and promoters.
15. The model (Figure 5) is not discussed in the text.
16. There are several typos, ej, GCIs (p4), Illunina (Methods)
17. The references in the methods (supplementary data) are not well cited, for instance: in the first page, references 29 and 32.

Point-by-point response to referees

Referee 1

The Hi-C data have good quality and are analyzed properly. The non-random DNA-DNA contacts are interesting and hint at potentially significant but unknown mechanisms for viral DNA targeting to the human genome. However, all presented data are correlative. No evidence is presented to support the model in figure 5. I have no problem with a speculative model but as presented, the authors give the impression that the data support their model. This is not the case. They have not shown that impeding viral-host genome contacts would adversely affect viral genome replication. They show that Cfp1 is required for effective HBV genome production but it is not clear if the DNA-DNA contacts play a role here.

We thank the reviewer for his comments and agree that the work was indeed to some extent correlative. To alleviate ambiguities and prevent misunderstandings with this respect we now insist that the model displayed in Figure 5 (now Figure 6) is a working hypothesis (page 9 and Fig. 6). The data do show that viruses present different strategies when it comes to spreading throughout the cellular genome organization and contact active chromatin. The active chromatin regions in contact with each of the viruses also do display different biological elements enriched in different transcription factors. We also show that Cfp1, one of the factors present at the CGI contacted by HBV, is required for viral transcription. Our working model is that each virus contacts different regions providing specific environments that are favorable to its own transcription/replication. Whether these repositioning are active or passive remain, indeed, unknown. In this revised version, we have started investigating this mechanism, although the protein we tested (HBx, see below), did not appear essential in this positioning.

We think that, although some of the observations are correlative, this work provides new findings and evidences that support exciting molecular studies of the mechanisms involved in this targeting and their functional consequences.

Why are there many more interactions between the Ad5 and host genomes than HBV-host? Are there differences in MOI or number of infecting particles? If so, would HBV show similar contact distribution if it was present at high copy number?

This difference is first due to genome length difference. Ad5 length is 35 kb while HBV is 3.2 kb. Because of this difference in size it is expected that more contacts will be caught for Ad5 than for HBV.

Whether the number of contacts would change by increasing the level of HBV DNA is impossible to answer directly because the level of cccDNA in the nucleus will never reach the level of Ad5 copies. The difference in the number of viral genomes after infection is due to the differences in the replication cycle of the two viruses. After infection, Ad5 amplifies its genome in the nucleus via DNA replication and the number of Ad5 genome can reach 10^4 , as confirmed in our experiments (supplementary Table 1) (Berk, Adenoviridae: The viruses and their Replication, Fields in Virology, Vol 2, Fifth Ed.). On the other hand, HBV doesn't carry a replication origin and doesn't replicate its genome by DNA replication in the nucleus. In fact, almost all of the HBV DNA found in nuclei after infection come from the DNA present in the viral particles during infection, and the pool of HBV cccDNA is not amplified during the course of infection to the extent of Ad5 DNA (Allweiss and Maura Dandri, viruses 2017). For each virus the level of viral DNA found in our experiments is consistent with former works. The level of cccDNA in infected PHH varies between 5 to 20 copies/cell depending on the studies (Riviere et al., J. Hepatol. 2015; Arzberger et al., J. Virol. 2010).

Nevertheless, we observed that the distributions of contacts detected for Ad5 at day 7, where approximately 6,000 to 9,000 viral genomes are present, are the same at day 4 when only 30 genomes are present, arguing that the number of viral genomes in the nucleus does not modify their positioning.

The authors indicate that “HBV contacts interfere with gene expression” in the abstract. This is just a conjecture and should be included only as part of the discussion. The same is true for Ad5.

We agree with the reviewer that this is a correlative observation and that the direct causality link is not demonstrated. We modified the abstract and we now discuss this point page 8. We nevertheless mention the correlation in the abstract, since it is a clinically relevant question: HBV nuclear DNA (cccDNA) persists in hepatocytes not only of chronically infected patients but also of patients who have spontaneously resolved the infection, and cccDNA persistence is associated with increased risk of developing hepatocellular carcinoma compared to non-infected patients.

On page 3, the authors indicate that Cfp1 regulates HBV transcription through H3K4me3. Again, the data is correlative and thus the statement is technically not correct.

Indeed we did not directly show the recruitment of Set1 and subsequent modification of H3K4me3. However, Cfp1 is required for H3K4me3 deposition and acts through the recruitment of Set1 complex containing histone methyltransferase Set1a or Set1b that deposit H3K4me3, allowing the establishment of an active chromatin state (Thomson et al., nature 2010, Brown et al., Cell reports, 2017). We also don't directly demonstrate that Cfp1 recruits Set1, but a study showed that Set1a is recruited on cccDNA and correlates with H3K4me3 (Alarcon et al. Scientific report 2016). Taken together, these data strongly point to mechanisms whereby Cfp1 would act through the recruitment of a methyltransferase such as Set1 that in turn deposits H3K4me3 on cccDNA.

Yet, to avoid over-interpretation, we modified the sentence on page 4 and wrote: "Altogether these results suggest that Cfp1 regulates HBV transcription through the recruitment of H3K4 methyltransferase allowing the deposition of H3K4me3 marks". In the discussion section, page 8 we also modified the text and discussed the possibility that Cfp1 is involved in the recruitment of Set1 for H3K4me3 as shown for cellular CGIs.

Please include lot numbers for the antibodies used in the paper in addition to the catalogue numbers.

We have included the batch numbers for all antibodies used in this work in Supplementary information, Methods, Antibodies and reagent section (HBc dako : lot 1018412 ; HBs , GR3199714-3, CGBP : lot # H0510, E1A : lot # G2611).

Referee 2

1. The major concern of the analysis is the resolution of the Hi-C and CHi-C. The resolution of the Hi-C is 400kbp, which is pretty low. The resolution of the CHi-C cannot be found in the manuscript or the supplementary materials. All the figures show the analysis results using 400kbp bins. At this resolution, it is unclear how to pinpoint the viral-host contacts. Within 400kbp genomic range, there can exist diverse functional regions including CpG islands,

promoters and enhancers. Or the viral DNAs are dozens of kbp, which allows pinpointing the viral-host interaction at a much higher resolution? Since this is the foundation of all the analyses, the details of determining viral-host contacts need to be provided.

Thanks for pointing this out, we realized that the resolution aspect was not clearly explained in the main text (and only available in supplementary information, Methods).

We used different resolutions depending on whether we performed analyses or display figures.

First, the chromosome contact maps displayed in Figure 1b, Figure 3b and supplementary Figures 2, 3, 9 and 10 are generated with a 400 kb resolution. The main reason is that it allows the reader to directly observe the TADs structure present in our data. The eigen vector computation to determine the partition in A/B compartment was also computed on 400 kb matrices. The contact profile of the virus was also displayed at this resolution to give the reader an overview of this signal. To sum up, low resolution contact maps (400 kb) were only used for visualization purpose (Figure 1B, C, D, E and Figure 3B, C, D, E).

Second, the statistical analyses were done using higher resolution contact data to determine the biological objects preferentially targeted by the virus genome. We fused Hi-C and Chi-C for the analysis of HBV and only Hi-C data for Ad5. Thus the resolution depends on the specific analysis performed:

- The overlap strategy and the histone enrichment analyses were done at the maximum resolution of our Hi-C experiments i.e. the restriction fragment level ($\pm 3,5$ kb around the start of the read) (Figures 2a, 2b, 2d, 3f, 5a and supplementary Figure 5).
- The enrichment plot analysis was done at a resolution of 50 kb (Figure 5b).
- The determination of the TADs borders was done using Hi-Cseg with the normalized contact data at 100 kb resolution.

Resolutions are now systematically indicated in the legend of figures.

2. Processing Hi-C and Chi-C data can be tricky. The supplementary method does not provide any critical detail such as normalization, correction of sequencing bias, call of contacts etc. It only mentioned to use a published SCN method to process the Hi-C data. How is this program compared to the other Hi-C processing packages? What are the parameters used for processing this data set?

We agreed with the referee that the process of Hi-C data can be tricky. Our lab (Kozul) hold a strong expertise in handling these data since their emergence in 2009/2010. We work with homemade programs that were developed over the past seven years by the lab, usually published as Methods sections of research articles (we never compiled everything into packages; see for instance Methods from Liroy et al., Cell, 2018; Muller et al., Mol. Sys. Biol., 2018; Mercy et al., Science, 2017; Lazar-Stefanita et al., EMBO Journal, 2017; Cournac et al., NAR, 2016 ; Lesne et al., Nature Methods, 2015, etc.) These analysis all requires extensive characterization and understanding of the contact data.

Notably, we published the SCN procedure in 2012 (Cournac et al., BMC Genomics) as a normalization algorithm very similar, but considerably faster, to the ICE normalization procedure published a bit afterwards by the Mirny lab in Nature Methods (with which the referee may be more familiar). This pioneering work also contains an in-depth description of the biases and unwanted/dubious ligation events that are present in a Hi-C library, providing solutions to alleviate them. SCN is a standard procedure that has been applied / cited on 90+ publications (400 times for the ICE, which is nearly identical). Codes developed for the different analyses are available on https://github.com/axelcournac/virus_Hi-C_Analysis.

We have expanded the Methods chapter to describe more precisely how the reads were handled, but we also think that the references we provide are sufficient to validate our computational approaches, that are not particularly original at this point, at least for us regarding our experience.

3. The change of TAD boundaries can be a very interesting observation after Ad5 infection. But the TAD boundary is around several kbp long. Given the 400kbp resolution of the host Hi-C data, how to determine the change of TAD boundary?

We apologize for the lack of clarity. The resolution used for the TADs analysis was 100 kb. Since the mean size of TADs is about 1 Mb (Dixon et al., Nature 2012; Ji et al., Cell Stem Cell 2016), this resolution allows to have a reasonable resolution regarding the characterization of TADs borders.

It was recently shown that Hi-Cseg algorithm gives more reproducible results (Mattia Forcato et al., Nature Methods 2017). See figure 1 below showing TADs border detection in PHH 345 day 4 post plating using Hi-Cseg.

Figure 1 : Example of TADs borders detection by Hi-CSeq , zoom of contact map of chromosome 12 (PHH345, day 4). Contact map was generated at 100kb resolution, each border is represented on the main diagonal by a orange dot.

Minor points:

1. The sequencing reads and depth of Hi-C and CHi-C are not described in Supplement.

Metrics regarding the sequencing reads are described in detail in supplementary table 1.

Referee 3

The manuscript is an elegant characterization of the sites where HBV and Ad5 genomes interact with the host genome. However, the work is descriptive and, before it can be considered for publication in Nature Communications, it would require mechanistic analysis.

We agree with the reviewer that a significant part of our data is correlative. We also would like to emphasize that this work nevertheless provides evidence that paves the way to future studies aiming at deciphering the mechanisms involved in the targeting of viral genomes and their functional consequences. Notwithstanding the correlative nature of the approach, we addressed several points that are, actually, mechanistically relevant. First, we show using CHi-C approaches that viruses contact preferentially active chromatin at CGIs for HBV and TSS and enhancer for Ad5 showing that these two viruses display specific localizations in the nucleus. Second, we show that Cfp1, a factor enriched at CGIs, is required for HBV transcription. We therefore show that each virus contacts different regions providing specific environments that are favorable to their own transcription/replication. Third, as requested by referee 3 question 9, we assessed the role of viral HBx in the targeting of HBV at CGIs. The results are now included in the manuscript pages 4 and 5, fig. 3 and Supplementary Table 1.

It is important to keep in mind that conformation chromosome capture techniques are relatively new approaches currently being used to study 3D genome organization and to understand the relationship between the architecture of the genome and cellular mechanisms such as cellular differentiation or cancer development. We want to underline that a correlative work is not synonymous of low impact work. Recent high-impact publications using Hi-C approaches featured mostly correlative studies bringing new and important insights on the role of transcription factors on chromatin state and genome topology changes that in turn lead to transcriptional changes during cell reprogramming (Stadhouders et al., Nature Genetics 2018,p238-249), or on the role of copy number variations or translocations occurring in cancer cells on 3D genome reorganization and gene regulation (Wu et al., Nature Communications 8, PMID: 29203764). These data, although correlative, also provide important foundations for future studies. Polymer or phase transition models can sometimes be used to support some theories accounting for the correlations, but still lacking molecular validation. A key point raised by referee 1 and referee 2 who mentioned that "Processing Hi-C and CHi-C data can be tricky » is to generate

high-quality data and robust analyses to settle the basis of future studies. This is what we intend to do in this work, which opens new perspectives and research hypothesis.

What is the consequence of the described interaction?

Our data show that the two studied DNA viruses, while contacting active chromatin, use different strategies to infiltrate the cellular genome 3D organization. The active chromatin regions in contact with each of the viruses display different biological elements enriched in different transcription factors. We showed that one of these factors present at the CGI contacted by HBV, Cfp1, is required for viral transcription. Using ChIP experiments we demonstrated that Cfp1 is recruited on the cccDNA and using siRNA, we showed that Cfp1 is required for HBV transcription since its depletion correlates with a decrease of HBV transcription and a decrease of H3K4me3 deposition. Cellular regions contacted by Ad5 are enriched for FOXA1 and FOXA2 and C/EBP binding sites that recruited factors known to be involved in Ad5 transcriptional regulation. We therefore suggest that each viruses contact different regions providing specific environments that are favorable to their own transcription/replication. This hypothesis will be tested in future work.

How is this interaction facilitated? Are there viral factors or cellular factors involved?

This question meets the question number 9 below, asking whether HBx could be involved in the targeting of HBV at CGIs.

Several mechanisms can be indeed envisioned, for instance a local and high concentration of cellular factors such as Cfp1 and additional transcriptional regulators may contribute to the recruitment of cccDNA (reminiscent of “phase transition” models that are gaining momentum in the field of 3D functional genome organization these days (Strom et al. 2017, Nature 547, p241). This mechanism was shown for the clustering of klf1 responsive genes, GCIs or interferon type 1 beta gene with chromatin region containing Nf-kB binding sites (Apostolou, E. & Thanos, D, *Cell* 134, 2008; Gushchanskaya et al., *Epigenetics* 9, 2014; Schoenfelder, S. et al., *Nat Genet* 42, 2010). Alternatively, as the reviewer suggests, the positioning of viral DNA at CGIs could also be triggered by a viral protein. HBx appears as a good candidate as it is involved in HBV transcriptional activity and is recruited by the cccDNA. We therefore tested this hypothesis and include the following result in the study.

To test whether HBx is involved in the virus positioning, we performed Hi-C and CHi-C using PHH (399) infected with HBV deficient for the expression of HBx (HBV X-), resulting in 116,502 contacts between the host and the HBV X- genome (Supplementary Table 1). In our experiments the genome organization of HBV wt or HBV X- infected hepatocytes appears very similar (see Fig. 3 of the revised manuscript). Interestingly, the HBV X- genome makes contacts with the entire host genome, with a significant enrichment for TSS and CpG islands similar to the one observed for the wild type virus (Figure 3c of the revised manuscript). Therefore, our data suggest that HBx silencing does not alter the positioning of the cccDNA.

We agree with the reviewer that understanding the mechanism leading to cccDNA positioning on the genome structure is important, and we started to characterize it by testing the HBV X- mutant. The present work, including the HB X- mutant, provides a solid ground for more experiments aiming at this goal. We think that testing more mutants and molecular mechanisms should be part of a future dedicated study, and we hope that the referee will agree.

The interactions observed are cause or consequences?

The contacts are real and their nature differs with the type of virus. Therefore, these contacts must relate to the metabolism of each virus and don't reflect the spontaneous repositioning of pieces of DNA in the nucleus. How does this targeting is orchestrated remains unknown, and under investigation. Again, the causality links in these 3D genomics studies are often difficult to establish, but our work provide a substantial number of elements to suggest that these contacts are functionally relevant, which is important in our opinion. We started investigating the role of the virus proteins in the maintenance of the contacts (below). However, this is a long-run objective, and we consider the present evidences are substantial enough to be reported.

The title does not reflect the findings of the manuscript: there is no indication that the interaction between viral and host genome affects both the host transcriptome and viral replication. It is not clear whether the interaction between viral and host genome affects the host transcription or whether this is a result of the viral expression. At the moment the manuscript only shows correlations.

We acknowledge the referee's point and this comment meets partially the third comment from referee 1. We agree and modified the abstract, and discuss this point. To avoid over-interpretation, we also modified the title.

Additional points:

1. References 2 and 3 in the abstract are incorrect.

We apologize for this mistake: references 2 and 3 now support the sentence "which likely favors the recruitment of Cfp1^{2,3}", as they support the idea that CpG islands are known to be bound by Cfp1 that in turn recruits histone methyltransferase complex.

2. Figure 1a and Supplementary Figure 1 show HBs staining in the uninfected PHH cells. At least two explanations can be proposed for this observation, the PHH cells derive from a chronically HBV infected patient, or, the HBs antibody is not-specific. This should be discussed.

We acknowledge the referee's point but a third explanation is proposed. It is known that hepatocytes and liver contain proteins with autofluorescence properties such as vitamin A, NAD(P)H (Croce et al., *Lasers in Surgery and medicine* 2010, Croce et al., *Photochem. Photobiol.* 2004).

As shown in Fig. 2a below, autofluorescence is detected in the green when PHH are observed under epifluorescence microscopy without any incubation with antibodies. The same picture is observed when non-infected PHH are incubated with anti-HBs and anti-HBc antibodies (Fig.2 b, below). This staining is quite different from the staining of HBV infected PHH with the anti-HBs antibodies which is more spread in the cytoplasmic (Fig.2 b below and Fig. 1a, enlarged picture in the manuscript).

To rule out the possibility that PHH are derived from chronically infected patient, we showed that non-infected PHH are negative for HBc staining (see Fig.2 below and Supplementary Fig.1 b in the revised manuscript) while infected PHH are positive for HBc staining.

We also assessed the expression of HBV RNA in non-infected and infected PHH by RT-qPCR using two sets of primers: one set that amplify the pgRNA and one set of primer that amplify all HBV transcripts (including the 2.4 and 2.1 kb RNA) excepted the 0,8 kb RNA encoding HBx. As shown in the manuscript in supplementary information Fig. 1a, no HBV transcripts are detected in non-infected PHH. Finally we do not detect cccDNA in non-infected PHH by qPCR (supplementary information Fig.1a). Altogether our results suggest that non-infected PHH do not contain HBV genome and the staining is due to autofluorescence of non-infected PHH.

Figure 2. PHH (399) were infected or not at a MOI of 500 viral particle (VP)/cell with HBV virus. **a)** non-infected PHH were directly observed by immunofluorescence microscopy using two different emissions filters (488nm, green and 647nm, far-red). **b)** HBV replication was assessed by immunofluorescence microscopy using antibody directed against HBs protein (green) and HBV core protein HBc (dako) (magenta). Mouse alexa fluor 488 was used for detection of HBs and rabbit alexa fluor 647 was used for detection of HBc. NI: non infected PHH. Coverslips were stained with DAPI for nuclear staining and then mounted with Vectashield (Vector Laboratories). a and b: Fluorescent images were acquired on an Axio Observer.Z1 microscope with an objective Plan-Neofluar 10x/0.30 Ph1. Images were acquired with AxioVision software (Carl Zeiss, Germany).

3. At the end of page 2, the authors mentioned that “At this resolution, the organization of the genome of hepatocytes appears similar to those of other cell types, whether the cells are infected or not”. However, the authors did not compare cell types, instead two different PHH are compared. Therefore, this needs to be corrected.

This point was unclear. We compared our Hi-C data generated on PHH with those generated on GM12878 cells from Rao and collaborators (Rao et al., Cell vol. 159, 2014). Data are shown in supplementary Fig. 2b, right diagram and supplementary Fig. 2c, left panel.

4. Figure 1F: please, explain the meaning of the orange and green colors.

Orange pics correspond to HBV contacts with active chromatin (type A), green pics correspond to HBV contacts with repressed chromatin (type-B). We modified the legend of figure 1F accordingly.

5. What percent of the total HBV interactions with host genome corresponds to CpG islands? This data will help to get an idea of the preference for the binding to CGI sites with respect to the whole interactions.

Fig.2a in the manuscript shows the proportion of contacts in % between HBV and a given element. 7% of all HBV contacts involve CpG islands. This percentage is compared to a random null model that shows the expected % of contacts at the biological elements.

The null models are computed to take into account the differences in detectability of genomic regions inherent to Hi-C protocols. These differences can be due to restriction sites density, GC content etc. [Yaffe et Tanay, 2010; Cournac et al., 2012; Cournact et al., 2016].

More precisely, for a group of N contacts between virus and human chromosomes, we generate a group of N random contacts by sampling with replacement the group of positions already detected in the whole Hi-C library. Doing that, a genomic region that is well detectable (and that will have more detected positions in the whole Hi-C library) will have more chance to be picked up and in contact with the virus.

We reiterate this procedure n times ($n = 1,000$ or $10,000$). From these n realizations, we can compute the expected distribution of proportion of contacts overlapping a biological group (as displayed with the violin plot). By comparing to the detected proportion of contact

involving the virus (HBV or Ad5), we can compute the p-value: for example, if we found 2 realizations among the $n = 10,000$ random sets with a proportion of contact superior or equal to the detected contact proportion for virus, the p-value is estimated to : $2 / 10,000 = 0.0002$.

The random groups are thus generated for each Hi-C library independently allowing to take into account the experimental variability in detection of each genomic region.

6. Figure 2C right (pathway enrichment of deregulated genes), the panel is not mentioned in the text.

Fixed. Page 4: KEGG pathway enrichment analysis showed significant enrichment in metabolic and liver-associated pathways such as complement and coagulation cascade, fructose and mannose metabolism or non-alcoholic fatty liver disease (Fig. 2c).

7. Figure 2D shows total HBV contacts in the context of biological elements (example: upregulated, and downregulated genes). It is also important to show, out of the 1139 mRNAs and 31 LncRNAs affected RNAs, what % of those genes actually contacts HBV.

Regarding the 1,139 deregulated mRNA, 16% of these genes contact at least one time HBV DNA. However calculating directly the % of genes contacted by HBV does not take into account the differences of detectability present in HI-C library and does not reflect the preferential contact frequency. This number gives only the general Hi-C detection rate of our experiments.

To properly assess the significance of contacts with different biological groups, we used a randomization procedure applied to groups of HBV contacts. We deliberately choose this analysis because doing the randomization procedure on the positions of elements from the biological groups would have been more complex to tackle due to the non uniform distribution of positions of those elements.

8. Figure 2E, western blots of siControl and siCfp1 (right panel) are not correctly labeled. In addition, the left panel, "ttx/roth" and "pg/roth" needs to be explained.

We are grateful to the referee for pointing this out, as indeed there were mistakes in the labelling, as he judiciously noticed! The ttx/roth and pg/roth in the left panel Fig. 2e was a typo accidentally left in the figure. The relative level of HBV RNA (total RNA or pgRNA) was

computed according to the ΔCt quantification method with $\Delta Ct = Ct \text{ HBV} - Ct \text{ Rhot2}$. Rhot2 was used as a reference gene because of its low variation coefficient in human liver tumors and cell lines. We corrected the mistake and modified the Fig.2 e accordingly.

9. Several groups, including this group, have shown that the viral protein HBx is important in regulating the chromatin state of cccDNA. Thus, how do HBx and Cfp1 relate in the HBV context? What would happen if the infection is performed with an HBV mutant defective for HBx expression (HBV X-)? Do the HBV and host genome interactions change under conditions where HBx is not present? Does HBx play any role in recruiting HBV to CGI regions? Does HBx interact with Cfp1? These questions could help in clarifying some of the mechanistic aspects missing in this manuscript.

We agree with the reviewer that it is important and exciting to uncover how HBV DNA is targeted at CGIs in active chromatin regions. Several mechanisms can be envisioned, for instance a local and high concentration of cellular factors such as Cfp1 and additional transcriptional regulators may contribute to the recruitment of cccDNA (reminiscent of “phase transition” models that are gaining momentum in the field of 3D functional genome organization these days (Strom et al. 2017, *Nature* 547, p241). This mechanism was shown for the clustering of klf1 responsive genes, GCIs or interferon type 1 beta gene with chromatin region containing Nf-kB binding sites (Apostolou, E. & Thanos, D, *Cell* 134, 2008; Gushchanskaya et al., *Epigenetics* 9, 2014; Schoenfelder, S. et al., *Nat Genet* 42, 2010). Alternatively, as the reviewer suggests, the positioning of viral DNA at CGIs could also be triggered by a viral protein. HBx appears as a good candidate as it is involved in HBV transcriptional activity and is recruited by the cccDNA. We therefore tested this hypothesis and include the following result in the study.

To test whether HBx is involved in the virus positioning, we performed Hi-C and Chi-C using PHH (399) infected with HBV deficient for the expression of HBx (HBV X-), resulting in 116,502 contacts between the host and the HBV X- genome (Supplementary Table 1). In our experiments the genome organization of HBV wt or HBV X- infected hepatocytes appears very similar (see Fig. 3 of the revised manuscript). Interestingly, the HBV X- genome makes contacts with the entire host genome, with a significant enrichment for TSS and CpG islands similar to the one observed for the wild type virus (Figure 3c of the revised

manuscript). Therefore, our data suggest that HBx silencing does not alter the positioning of the cccDNA.

We agree with the reviewer that understanding the mechanism leading to cccDNA positioning on the genome structure is important, and we started to characterize it by testing the HBV X- mutant. The present work, including the HB X- mutant, provides a solid ground for more experiments aiming at this goal. We think that testing more mutants and molecular mechanisms should be part of a future dedicated study, and we hope that the referee will agree.

We therefore proposed different mechanisms that could be involved in the positioning of viral DNA at active chromatin including the potential role of viral proteins in this process in the discussion page 7.

We also must point out that while (10 days ago) we were completing the revised version of this work, a study was published that seemingly contradict the above results (Hensel et al., *Epigenetics & Chromatin*, (2018) 11:34). In this work the authors claim that HBx is involved in the positioning of HBV DNA to active chromatin, something our experiment does not support. We immediately downloaded their data (SRA library 4C_cccDNA-HepG2.2.15) and proceeded to their analysis, while examining their experimental proceeding. Surprisingly, we could not find any results in the work supporting this claim. In addition, we noticed what we consider as flaws in their data and experiments that concern us regarding the significance of this paper, as we doubt that their chromosome conformation capture (4C) experiment contains any meaningful contact signal.

First, the HepG2.2.15 cells used in this work carry genomic integrations of HBV (Watanabe et al. *Genome Res.* 2015 Mar; 25(3): 328–337). A 4C capture experiment using HBV sequence as a prey performed on these cells should therefore lead to enrichment in contacts in the regions flanking these integration sites and exponentially decreasing in both directions. This enrichment at the five known HBV integration sites was not present in the data (Figure 3 below). What was however very clear was a strong correlation between the 4C coverage and the density of restriction sites *HaeIII* (GGCC) used to perform the second restriction of the 4C experiment (Figure 4 below).

Figure 3: Zoom of chromosome 4 showing the 4C coverage (blue) and the number of restriction sites (HaeIII) per 50 kb bins. The known integration of HBV genome in chromosome 4 is shown with orange dot and dashed line. Signals are Z-transformed for visualization purpose.

Figure 4: Density plot showing number of restriction sites (HaeIII) per 50 kb bins versus the 4C coverage. Each point represents one 50 kb bin and the scatter plot is realized on the whole genome. Pearson coefficient is also indicated.

This high correlation ($PC \sim 0.4$) found between their contacts using an enzyme that targets preferentially GC rich region, and the read coverage of these GC rich region strongly suggest that this is the signal they interpret as biologically relevant without using a proper null model to test for this skewed coverage. As these genomic regions are enriched in genes

the claim that the virus contacts active regions, even if true, is not supported by the data they provide. Any experiment involving a HaeIII and not properly normalized would lead to the same result.

Besides these technical concerns, we also think the authors over-interpret their data. The claim that HBx is involved in HBV DNA positioning is based on a comparison between the genomic DNA regions enriched in HBx in a ChIP-seq experiment performed using anti-HBx antibodies, and genomic DNA regions enriched in contact with HBV. They don't use a virus defective for the expression of HBx to perform a 4C analysis! It is really surprising that the authors were let to conclude that HBx drive the targeting, as there is no way their experiment allows any conclusion (besides showing that HBx co-localize with HBV!)

We decided to not include this discussion in the revised version of our work and to ignore this very recent literature that, in our opinion, only adds noise to the field and is not really relevant.

10. It is not clear to me what the authors wants to say with "PHH donor (345) display a natural infection by the human adenoviruses serotype 5...". The other PHH cells cannot be infected with Ad5? Please, clarify this.

PHH 345 derived from a 9 months old baby that appears to be infected by Ad5. We did not perform thus the infection with Ad5 in the laboratory but take advantage of these PHH infected with Ad5 to study the nuclear localization of the virus. We believe that the PHH were already infected rather than a contamination that could occurred in the laboratory, since we used different vials of PHH 345, plated them at different days and, recurrently, we observed the same profile of infection with all cells in the culture being infected at day 7 post plating. As shown in supplementary Fig. 11 in the revised manuscript, we carefully followed Ad5 infection by measuring the expression of early viral RNAs (E1A, E1B) and late viral RNA (Hexon) as well as viral DNA replication. We also follow E1A expression by immunofluorescence and western blot (supplementary Fig.11 in the revised manuscript). Interestingly we estimated the number of viral genomes at days 7 post plating around 6 000 to 9 100, in accordance with the literature (Berk, Adenoviridae: The viruses and their Replication, Fields in Virology, vol 2, Fifth Ed.). Moreover using RT-qPCR, we confirmed in our setting, the up regulation of cellular genes (such as CCNE2, CCND2) known to be up regulated during Ad5 infection.

We modified the sentence in the text page 5 to clearly indicate that PHH 345 are derived from a donor already infected by Ad5.

11. There is no description regarding the PHH infection with Ad5 as well as how the Ad5 virus was produced.

As mentioned before, PHH 345 derived from a 9 months old baby that was infected by Ad5.

12. The presentation of the Ad5 data is confusing, going back and forward from Figure 3 to Figure 4. This should be corrected.

We acknowledge the referee's comment and agree that the sentence: "Intriguingly, a small enrichment between Ad5 and regions displaying the repressive H3K27me3 mark was also observed at day 7 p.p., but not 4 p.p. (Fig. 4h in the revised manuscript, former Fig. 3h), raising the possibility that those regions may have shifted to compartment A, an hypothesis which remains to be tested" has to be regrouped with the part of the text describing the figure Fig. 4h. We modified the text accordingly page 6.

13. Figure 3C is not discussed in the text.

Thanks for pointing this out, this is fixed page 5. Fig.4c (former figure 3c) shows the distribution of H3K4me3 along the chromosome 12 using dataset from the Encode project generated. Fig. 4d (former Figure 3d) corresponds to the eigen vector decomposition. Analyses allow the partitioning of the genome into A (active) and B (inactive) compartment.

14. In page 5, second paragraph, it is mentioned that "...Ad5 contacted preferentially regions enriched for active marks at promoters and enhances..." and refers to Figure 3h. However, this figure only shows enrichment on histone modifications globally, without looking at enhancers and promoters.

We agree with the reviewer. What we meant was that Ad5 contacts regions enriched for active marks (i.e. H3K4me3 and H3K27 ac) known to mark active promoters and enhancers. We modified the text accordingly page 6.

15. The model (Figure 5) is not discussed in the text.

We acknowledge the referee's comment and discussed the model in the discussion page 9. The model is now shown in Figure 6 (former Fig. 5).

16. There are several typos, ej, GCIs (p4), Illunina (Methods)

We apologize for the mistakes and we have corrected the errors.

17. The references in the methods (supplementary data) are not well cited, for instance: in the first page, references 29 and 32.

We thank the reviewer for his comment. We added the references in the supplementary information in Cell culture, HBV production and infection section and in Supplementary References section (page 10).

Reviewers' comments:

Reviewer #1 (Remarks to the Author):

The authors have partially addressed my concerns by softening some of the statements. But I think they have not gone far enough. For instance, the abstract poses the question of "Whether DNA viruses take advantage of the higher-order architecture of mammalian genomes to recruit or mobilize factors necessary to their own replication remains largely unknown." Normally, one would expect the paper to provide the answer to this question. But the manuscript remains correlative, and provides no evidence to support such implied claim.

I agree that it has been unclear whether viral genomes make non-random contacts with cellular genomes. This paper provides strong evidence for non-random contacts. I think the authors should consider presenting their argument from this perspective. There may be many reasons for such non-random interactions, one of which is what the authors suggest. But it may turn out that there are other explanations. I think it would be sufficient if the authors re-wrote their abstract so as not to raise the expectation that they would show mobilization of transcription factors at the points of contact.

A paper on bioRxiv (doi: <https://doi.org/10.1101/142604>) has also examined the interaction of Ad5 genome with cellular genome in fibroblasts. I don't know what the standard is for citing bioRxiv papers but it would be interesting to compare the two findings.

Reviewer #2 (Remarks to the Author):

The authors clarified that the TAD borders was called at 100kbp resolution. But Fig. 1 in the response is not convincing that the called TAD borders were consistent with visual inspection. It can be due to different reasons and resolution may still be a confounding factor.

Reviewer #3 (Remarks to the Author):

The manuscript entitled "Tridimensional infiltration of DNA viruses into the host genome: implication for gene expression and viral replication" by the authors Moreau et al. is a revised version. The manuscript is significantly improved over its previous version. The authors have satisfactorily addressed all questions raised by this referee. In this revised version, the authors clarified many things that were unclear, changed the title to fit more with their results, they gave more details in result section, material and methods, figure legends. In addition, they included new results (Figure 3 and Figure 5c). In all, the manuscript can be accepted for publication.

Point-by-point response to referees

Reviewer #1 (Remarks to the Author):

The authors have partially addressed my concerns by softening some of the statements. But I think they have not gone far enough. For instance, the abstract poses the question of "Whether DNA viruses take advantage of the higher-order architecture of mammalian genomes to recruit or mobilize factors necessary to their own replication remains largely unknown." Normally, one would expect the paper to provide the answer to this question. But the manuscript remains correlative, and provides no evidence to support such implied claim.

I agree that it has been unclear whether viral genomes make non-random contacts with cellular genomes. This paper provides strong evidence for non-random contacts. I think the authors should consider presenting their argument from this perspective. There may be many reasons for such non-random interactions, one of which is what the authors suggest. But it may turn out that there are other explanations. I think it would be sufficient if the authors re-wrote their abstract so as not to raise the expectation that they would show mobilization of transcription factors at the points of contact.

We now have modified the abstract accordingly. We kept a final sentence that raise the hypothesis that targeting these regions may favor the recruitment of TF. This is not an uncommon thing to do and we hope the referee will agree this is appropriate in the present case.

« During infection, non-integrated viral DNA stands in the vicinity of the genome of the host cell. Whether DNA viruses distribute randomly in the nuclear space or target specific positions within the higher-order architecture of mammalian genomes remains largely unknown. We used Hi-C and viral DNA capture (Chi-C) in primary human hepatocytes (PHH) infected by either hepatitis B virus (HBV) or adenovirus type 5 (Ad5) virus to show that they adopt different strategies in their respective positioning at active chromatin. HBV contacts preferentially CpG islands (CGIs), which may favors the recruitment of Cfp1, a factor required for its transcription. The targeted CGIs are often associated with genes highly expressed in PHH and deregulated during infection. On the other hand, Ad5 DNA contacts preferentially transcription start sites (TSSs) and enhancers of highly expressed genes and genes up-regulated during infection. These results show that DNA viruses use different strategies to infiltrate genomic 3D networks and target specific regions. This targeting may facilitate the recruitment of transcription factors necessary for their own replication, and as a result contribute to the deregulation of cellular gene expression. »

A paper on bioRxiv (doi: <https://doi.org/10.1101/142604>) has also examined the interaction of Ad5 genome with cellular genome in fibroblasts. I don't know what the standard is for citing bioRxiv papers but it would be interesting to compare the two findings.

The raw reads of this work are not accessible online, making it difficult to compare with our own data. Overall, the conclusions are quite similar.

Reviewer #2 (Remarks to the Author):

The authors clarified that the TAD borders was called at 100kbp resolution. But Fig. 1 in the response is not convincing that the called TAD borders were consistent with visual inspection. It can be due to different reasons and resolution may still be an confounding factor.

We now discuss the limitation of the data in the detection of TAD boundaries (and refer more explicitly with the same analysis performed using data from Rao et al., 2015).

Reviewer #3 (Remarks to the Author):

The manuscript entitled “Tridimensional infiltration of DNA viruses into the host genome: implication for gene expression and viral replication” by the authors Moreau et al. is a revised version. The manuscript is significantly improved over its previous version. The authors have satisfactorily addressed all questions raised by this referee. In this revised version, the authors clarified many things that were unclear, changed the title to fit more with their results, they gave more details in result section, material and methods, figure legends. In addition, they included new results (Figure 3 and Figure 5c). In all, the manuscript can be accepted for publication.

We thank the referee for his appreciation of our revised work.